# Structures and Bioactivities of Six New Triterpene Glycosides, Psolusosides E, F, G, H, H_1_, and I and the Corrected Structure of Psolusoside B from the Sea Cucumber *Psolus fabricii*

**DOI:** 10.3390/md17060358

**Published:** 2019-06-14

**Authors:** Alexandra S. Silchenko, Anatoly I. Kalinovsky, Sergey A. Avilov, Vladimir I. Kalinin, Pelageya V. Andrijaschenko, Pavel S. Dmitrenok, Roman S. Popov, Ekaterina A. Chingizova, Svetlana P. Ermakova, Olesya S. Malyarenko

**Affiliations:** G.B. Elyakov Pacific Institute of Bioorganic Chemistry, Far Eastern Branch of the Russian Academy of Sciences, Pr. 100-letya Vladivostoka 159, Vladivostok 690022, Russia; sialexandra@mail.ru (A.S.S.); kaaniv@pidoc.dvo.ru (A.I.K.); avilov-1957@mail.ru (S.A.A.); pandryashchenko@mail.ru (P.V.A.); paveldmt@piboc.dvo.ru (P.S.D.); rs.popov@outlook.com (R.S.P.); martyyas@mail.ru (E.A.C.); svetlana_ermakova@hotmail.com (S.P.E.); malyarenko.os@gmail.com (O.S.M.)

**Keywords:** *Psolus fabricii*, triterpene glycosides, psolusosides, sea cucumber, cytotoxic activity

## Abstract

Seven sulfated triterpene glycosides, psolusosides B (**1**), E (**2**), F (**3**), G (**4**), H (**5**), H_1_ (**6**), and I (**7**), along with earlier known psolusoside A and colochiroside D have been isolated from the sea cucumber *Psolus fabricii* collected in the Sea of Okhotsk. Herein, the structure of psolusoside B (**1**), elucidated by us in 1989 as a monosulfated tetraoside, has been revised with application of modern NMR and particularly MS data and proved to be a disulfated tetraoside. The structures of other glycosides were elucidated by 2D NMR spectroscopy and HR-ESI mass-spectrometry. Psolusosides E (**2**), F (**3**), and G (**4**) contain holostane aglycones identical to each other and differ in their sugar compositions and the quantity and position of sulfate groups in linear tetrasaccharide carbohydrate moieties. Psolusosides H (**5**) and H_1_ (**6**) are characterized by an unusual sulfated trisaccharide carbohydrate moiety with the glucose as the second sugar unit. Psolusoside I (**7**) has an unprecedented branched tetrasaccharide disulfated carbohydrate moiety with the xylose unit in the second position of the chain. The cytotoxic activities of the compounds **2**–**7** against several mouse cell lines—ascite form of Ehrlich carcinoma, neuroblastoma Neuro 2A, normal epithelial JB-6 cells, and erythrocytes—were quite different, at that hemolytic effects of the tested compounds were higher than their cytotoxicity against other cells, especially against the ascites of Ehrlich carcinoma. Interestingly, psolusoside G (**4**) was not cytotoxic against normal JB-6 cells but demonstrated high activity against Neuro 2A cells. The cytotoxic activity against human colorectal adenocarcinoma HT-29 cells and the influence on the colony formation and growth of HT-29 cells of compounds **1**–**3**, **5**–**7** and psolusoside A was checked. The highest inhibitory activities were demonstrated by psolusosides E (**2**) and F (**3**).

## 1. Introduction

The sea cucumbers triterpene glycosides are long-time investigated natural compounds characterized by significant structural diversity, exhibiting a broad spectrum of biological activity [1,2,3,4,5,6,7,8,9]. Some of them are under study as marine drugs.

The investigation of a complicated glycoside composition of the sea cucumber *Psolus fabricii* (Psolidae, Dendrochirotida) was started in the 1980s of XX century. Only two main compounds, psolusosides A [10,11] and B [12,13], had been isolated in that time. Recently, we have recommenced the studies on the glycosides of *P. fabricii* that resulted in the isolation of eight new hexaosides, psolusosides C_1_–C_3_ and D_1_–D_5_, as well as five previously known compounds [14,15]. Herein, we report the isolation and structural elucidation of six new glycosides, psolusosides E (**2**), F (**3**), G (**4**), H (**5**), H_1_ (**6**), and I (**7**), as well as an earlier known psolusoside B (**1**)**,** whose structure has been revised based on the modern NMR and HR MS techniques. Earlier known glycosides, psolusoside A and colochiroside D, were also isolated and identified. The structures of the glycosides were established based on ^1^H, ^13^C NMR, and 1D TOCSY spectra and 2D NMR (^1^H,^1^H-COSY, HMBC, HSQC, ROESY) and confirmed by HR-ESI mass spectrometry. The hemolytic activities against mouse erythrocytes and cytotoxic activities against mouse Ehrlich carcinoma cells (ascite form), neuroblastoma Neuro 2A cells and normal epithelial JB-6 cells of **2**–**7** have been studied. Psolusoside I (**7**) demonstrated moderate hemolytic activity when compounds **2**–**6** were highly hemolytic, but none of them, with the exception of known psolusoside A, which was used as control, were not cytotoxic against mouse Ehrlich carcinoma cells. Psolusoside G (**4**) was not cytotoxic against normal JB-6 cells but demonstrated high activity against Neuro 2A cells. Psolusosides E (**2**) and F (**3**), with the holostane aglycones and linear tetrasaccharide monosulfated sugar chains, demonstrated the highest in the series of tested compounds inhibitory activity on the colony formation and growth of H-29 cells.

## 2. Results and Discussion

### 2.1. Structural Elucidation of the Glycosides

The concentrated ethanolic extract of *P. fabricii* was re-extracted with СHCl_3_/MeOH, concentrated, and delipidized with EtOAc/H_2_O. The water layer was chromatographed on a Polychrom-1 (powdered Teflon, Biolar, Latvia) in 50% EtOH and on Si gel columns using CHCl_3_/EtOH/H_2_O (100:75:10), (100:100:17) and (100:125:25) as mobile phases to give fractions I–VIII. The obtained fractions III–VIII were subjected to HPLC on reversed-phase or silica-based columns to give psolusosides: B (**1**) (67 mg), E (**2**) (10 mg), F (**3**) (1.4 mg), G (**4**) (46.5 mg), H (**5**) (1.4 mg), H_1_ (**6**) (1.4 mg), and I (**7**) (1.1 mg) (Figure 1) as well as two known earlier compounds, psolusoside A (36.5 mg) found earlier in this species of sea cucumbers [10,11] and colochiroside D (2.5 mg) isolated first from *Colochirus robustus* [16]. The known compounds were identified by comparison of their ^1^H and ^13^C NMR spectra with those reported for psolusoside A (3β-*O*-[6-*O*-sodium sulfate-3-*O*-methyl-β-d-glucopyranosyl-(1→3)-6-*O*-sodium-sulfate-β-d-glucopyranosyl-(1→4)-β-d-quinovopyranosyl-(1→2)-β-d-xylopyranosyl]-16-ketoholosta-9(11),25-diene) and colochiroside D (3β-*O*-[3-*O*-methyl-β-d-glucopyranosyl-(1→3)-6-*O*-sodium-sulfate-β-d-glucopyranosyl-(1→4)-β-d-glucopyranosyl-(1→2)-β-d-xylopyranosyl]-16-ketoholosta-9(11),25-diene).

The structure of psolusoside B assigned earlier [12,13] was shown to be monosulfated branched tetraoside with non-holostane aglycone, namely 3β-*O*-{β-d-glucopyranosyl-(1→4)-β-d-glucopyranosyl-(1→2)-[6-*O*-sodium-sulfate-β-d-glucopyranosyl-(1→4)]-β-d-xylopyranosyl}-9βH,20(*S*)-acetoxylanosta-7,25-diene-18(16)-lactone.

However, the reinvestigation has shown that this glycoside has two sulfate groups instead of the one reported earlier. In fact, the more accurate molecular formula of psolusoside B (**1**) was determined to be C_55_H_84_O_30_S_2_Na_2_ from the [M_2Na_ + Na]**^+^** ion peak at *m/z* 1357.4169 (calc. 1357.4176) and [M_2Na_ + 2Na]^2+^ at *m/z* 690.2039 (calc. 690.2034) in the (+)HR-ESI-MS and indicated the presence of two sulfate groups in **1**. The comparison of 1D and 2D NMR spectra of the aglycone part of psolusoside B (**1**) (Table 1, Appendix A) with those of the aglycone part of colochiroside E, isolated from the sea cucumber *Colochirus robustus* [17], has confirmed their identity with the aglycone of psolusoside B elucidated earlier [12]. Thus, psolusoside B (**1**), isolated by us, actually contains non-holostane aglycone with 18(16)-lactone and *O*-acetic group at C-20, which was described earlier as onekotanogenin.

In the ^1^H and ^13^C NMR spectra (Table 2, Appendix A) of the carbohydrate part of **1,** four characteristic doublets at δ (H) 4.56–5.11 (*J* = 7.3–7.9 Hz) and, corresponding to them, four signals of anomeric carbons at δ(C) 100.9–104.8 (Appendix A) were indicative of a tetrasaccharide chain and *β*-configurations of glycosidic bonds. The ^1^H,^1^H-COSY and 1D TOCSY spectra of **1** showed the signals of the isolated spin systems assigned to one xylose and three glucose residues (Appendix A). The positions of interglycosidic linkages were established by the ROESY and HMBC spectra of **1** (Table 2, Appendix A) where the cross-peaks between H(1) of the xylose and H(3) (C(3)) of an aglycone, H(1) of the second residue (glucose) and H(2) (C(2)) of the xylose, H(1) of the third residue (glucose) and H(4) (C(4)) of the second residue (glucose), and H(1) of the fourth residue (glucose) and H-4 (C(4)) of the first residue (xylose), were observed. These data indicated the same architecture (tetrasaccharide branched chain) and monosaccharide composition of sugar chain of **1** as it has been reported earlier [13]. Thorough analysis of the NMR spectra of **1** showed the glucose residue (the third sugar unit, in which signals were deduced by ^1^H,^1^H-COSY, and confirmed by 1D TOCSY) attached to C(4) of the second sugar unit (glucose) was sulfated by C(6) due to α- and β-shifting effects observed in the ^13^C NMR spectrum. Really, the signal of C(6) was observed at δ(C) 67.5 and the signal of C(5) at δ(C) 75.5. Hence, these signals were shifted in comparison with corresponding signals (δC_(6)_ at 62.1, δC_(5)_ at 77.8) in non-sulfated glucose residue in the same position of carbohydrate chains of psolusosides, belonging to the group D [15]. The analogous shifting effects were observed for the signals C(2) and C(1) of the glucose occupying the fourth position of the carbohydrate chain of **1** and attached to C(4) of the xylose unit. The signal of C(2) of this residue was observed at δ 80.6 due to the attachment of a sulfate group to this position while the signal of C(1) was shifted upfield to δ(C) 100.9 due to β-effect of a sulfate group. Moreover, the signals in the ^13^C NMR spectrum of **1** assigning to this glucose residue were coincident with the corresponding signals of glucose residue sulfated by C-2 and attached to C-4 of the first xylose unit in the spectrum of colochiroside E [17] corroborating the unusual position of one of sulfate groups in psolusoside B (**1**). So, the both spectroscopic methods—HR-ESI-MS and NMR—confirmed the presence of two sulfate groups in the carbohydrate chain of psolusoside B (**1**).

The structure of **1** was also confirmed by the (+)ESI-MS/MS of the [M_2Na_ + Na]^+^ ion at *m/z* 1357.4, in which the peaks of fragment ions were observed at *m/z* 1297.4 [M_2Na_ + Na − CH_3_COOH]^+^, 1237.4 [M_2Na_ + Na − NaHSO_4_]^+^, 1177.4 [M_2Na_ + Na − CH_3_COOH − NaHSO_4_]^+^, 1117.4 [M_2Na_ + Na − 2NaHSO_4_]^+^, 1075.5 [M_2Na_ + Na − C_6_H_10_O_9_SNa (GlcSO_3_Na)]^+^, 913.4 [M_2Na_ + Na − GlcSO_3_Na − Glc]^+^, 863.1 [M_2Na_ + Na − C_32_H_47_O_4_ (Agl) + H]^+^, 743.1 [M_2Na_ + Na − C_32_H_47_O_4_ (Agl) − NaHSO_4_]^+^, 581.1 [M_2Na_ + Na − C_32_H_47_O_4_ (Agl) − C_6_H_10_O_9_SNa (GlcSO_3_Na)]^+^, 449.0 [M_2Na_ + Na − C_32_H_47_O_4_ (Agl) − C_6_H_10_O_9_SNa (GlcSO_3_Na) − Xyl]^+^, 287.0 [M_2Na_ + Na − C_32_H_47_O_4_ (Agl) − C_6_H_10_O_9_SNa (GlcSO_3_Na) − Xyl − Glc]^+^.

Based on these results, the structure of psolusoside B (**1**) was determined as 3β-*O*-{6-*O*-sodium sulfate-β-d-glucopyranosyl-(1→4)-β-d-glucopyranosyl-(1→2)-[2-*O*-sodium sulfate-β-d-glucopyranosyl-(1→4)]-β-d-xylopyranosyl}-9βH,20(*S*)-acetoxylanosta-7,25-diene-18(16)-lactone.

Colochiroside E [17], having trisaccharide sugar chain with terminal (glucose) residue sulfated by C(2), differed from psolusoside B (**1**) only by the lack of a terminal glucose residue attached to C(4) of the glucose (the second unit in the chain). This fact indicates the biogenetic interconnection of these compounds: colochiroside E seems to be a biosynthetic precursor of psolusoside B (**1**) that, additionally, corroborates the new structure of **1**. The incorrect structure elucidation of psolusoside B in 1989 [13] could be explained by an ambiguity of interpretation of the ^13^C NMR signals. The use of FAB-MS for the molecular formula calculation obviously resulted in the desulfation of the glycoside during the spectrum registration.

The ^1^H and ^13^C NMR spectra of aglycone parts of psolusosides E (**2**), F (**3**), and G (**4**) were coincident to each other showing the identity of the aglycones in these glycosides. In the aglycone part of the ^13^C NMR spectra of **2**–**4,** the signals characteristic of 18(20)-lactone (δ(C) 175.8 C(18) and 82.9 (C(20)), 9(11)- (δ(C) 151.2 C(9) and 110.9 C(11)), and 25(26)-double bonds (δ(C) 145.4 C(25) and 110.3 C(26)), as well as the signal of C-16 keto-group (δ(C) 212.9) were observed (Table 3, Appendix A). Based on the analysis of the NMR spectra, the aglycone of compounds **2**–**4** was identified as earlier known 16-ketoholosta-9(11),25-dien-3β-ol, found first in holotoxins A_1_ and B_1_ from the sea cucumbers belonging to the family Stichopodidae [18], and frequently occurred in the glycosides of different sea cucumber taxa.

The molecular formula of psolusoside E (**2**) was determined to be C_54_H_83_O_25_SNa from the [M_Na_ − Na]**^−^** ion peak at *m/z* 1163.4945 (calc. 1163.4950) in the (−)HR-ESI-MS. In the ^1^H and ^13^C NMR spectra of the carbohydrate part of psolusoside E (**2**), four characteristic doublets at δ(H) 4.77–5.22 (*J* = 7.3–7.7 Hz) and, corresponding to them, signals of anomeric carbons at δ(C) 104.6–105.4 were indicative of a tetrasaccharide chain and β-configurations of glycosidic bonds (Table 4, Appendix A). The ^1^H,^1^H-COSY, and 1D TOCSY spectra of **2** showed the signals of four isolated spin systems assigned to the xylose, quinovose, glucose, and 3-*O*-methylglucose residues (Appendix A). The positions of interglycosidic linkages were elucidated by the ROESY and HMBC spectra of **2** (Table 4, Appendix A) by same manner as for **1,** indicating the presence of linear tetrasaccharide chain in psolusoside E (**2**). The signals of C(6) and C(5) of the glucose residue (the third sugar), observed at δ(C) 67.6 and 75.0, correspondingly, were characteristic of the sulfated by C(6) glucopyranose residue. Thus, psolusoside E (**2**) is a monosulfated tetraoside, with the glucose residue, sulfated by C(6), as the third monosaccharide unit. Such carbohydrate chain was not found earlier in the glycosides from sea cucumbers.

The (−)ESI-MS/MS of **2** demonstrated the fragmentation of [M_Na_ − Na]^−^ ion at *m/z* 1163.5. The peaks of fragment ions were observed at *m/z*: 987.4 [M_Na_ − Na − MeGlc + H]^−^, 695.2 [M_Na_ − Na − C_30_H_43_O_4_ (Agl) − H]^−^, 563.1 [M_Na_ − Na − C_30_H_43_O_4_ (Agl) − Xyl]^−^, 417.1 [M_Na_ − Na − C_30_H_43_O_4_ (Agl) − Xyl − Qui]^−^, 241.0 [M_Na_ − Na − C_30_H_43_O_4_ (Agl) − Xyl − Qui − MeGlc]^−^ (corresponds to desodiated sulfated glucose residue) corroborating the structure of psolusoside E (**2**).

All these data indicate that psolusoside E (**2**) is 3β-*O*-[3-*O*-methyl-β-d-glucopyranosyl-(1→3)-6-*O*-sodium-sulfate-β-d-glucopyranosyl-(1→4)-β-d-quinovopyranosyl-(1→2)-β-d-xylopyranosyl]-16-ketoholosta-9(11),25-diene.

The molecular formula of psolusoside F (**3**) was determined to be C_54_H_83_O_25_SNa from the [M_Na_ − Na]**^−^** ion peak at *m/z* 1163.4952 (calc. 1163.4950) in the (−)HR-ESI-MS and was coincident with the formula of psolusoside E (**2**). In the ^1^H and ^13^C NMR spectra of the carbohydrate part of psolusoside F (**3**), four characteristic doublets at δ(H) 4.71–5.12 (*J* = 7.3–8.2 Hz) and corresponding signals of anomeric carbons at δ(C) 104.0–105.0 were indicative of a tetrasaccharide chain and *β*-configurations of glycosidic bonds (Appendix A). The positions of interglycosidic linkages were elucidated by the ROESY and HMBC spectra of **3** (Table 5, Appendix A) as described above, indicating the presence of linear tetrasaccharide carbohydrate chain. The monosaccharide composition of **3**, deduced from the ^1^H,^1^H-COSY, and 1D TOCSY spectra (Appendix A), was the same as in **2**. The comparison of the ^13^C NMR spectra of these compounds showed the coincidence of their signals corresponding to xylose and quinovose residues. The signals of C(6) and C(5) of the glucose residue (the third unit in the chain) in the ^13^C NMR spectrum of **3** were observed at δ(C) 61.7 (shielded as compared with corresponding signal in the spectrum of **2**) and 77.1 (de-shielded as compared with C(5) of the glucose in the spectrum of **2**), correspondingly, indicating the absence of a sulfate group in this residue. The signal of C(6) of 3-*O*-methyl-glucose residue was observed at δ(C) 67.0 and the signal C(5) of the same residue—at δ(C) 75.6 in the ^13^C NMR spectrum of **3** indicating the attachment of a sulfate group to C(6) of terminal 3-*O*-methyl-glucose unit in the carbohydrate chain of psolusoside F (**3**). So, psolusosides E (**2**) and F (**3**) differed from each other only in the position of a sulfate group. The carbohydrate chain of **3** is a new one.

The (−)ESI-MS/MS of **3** demonstrated the fragmentation of [M_Na_ − Na]^−^ ion at *m/z* 1163.5. The peaks of fragment ions were observed at *m/z*: 695.2 [M_Na_ − Na − C_30_H_43_O_4_ (Agl) − H]^−^, 563.1 [M_Na_ − Na − C_30_H_43_O_4_ (Agl) − Xyl]^−^, 417.1 [M_Na_ − Na − C_30_H_43_O_4_ (Agl) − Xyl − Qui]^−^, 255.0 [M_Na_ − Na – C_30_H_43_O_4_ (Agl) – Xyl – Qui – Glc]^−^, confirming the sequence of monosaccharides in the sugar chain.

All these data indicate that psolusoside F (**3**) is 3β-*O*-[6-*O*-sodium-sulfate-3-*O*-methyl-β-d-glucopyranosyl-(1→3)-β-d-glucopyranosyl-(1→4)-β-d-quinovopyranosyl-(1→2)-β-d-xylopyranosyl]-16-ketoholosta-9(11),25-diene.

The molecular formula of psolusoside G (**4**) was determined to be C_54_H_82_O_29_S_2_Na_2_ from the [M_2Na_ − Na]**^−^** ion peak at *m/z* 1281.4313 (calc. 1281.4286) in the (−)HR-ESI-MS indicating the presence of two sulfate groups in this glycoside. In the ^1^H and ^13^C NMR spectra of the carbohydrate part of psolusoside G (**4**), four characteristic doublets at δ(H) 4.72–5.16 (*J* = 7.2–8.4 Hz) and, corresponding to them, signals of anomeric carbons at δ(C) 103.8–105.0 were indicative of a tetrasaccharide chain and β-configurations of glycosidic bonds (Table 6, Appendix A).

Analysis of the ^1^H,^1^H-COSY and 1D TOCSY spectra of psolusoside G (**4**) showed the availability of one xylose, two glucose, and one 3-O-methyl-glucose residues (Appendix A). So, the quinovose unit was absent in the chain of **4** that was corroborated by the lack of characteristic doublet of methyl group of quinovose residue at δ(H) ≈1.70 in the ^1^H NMR spectrum and the corresponding signal at δ(C) ≈18.0 in the ^13^C NMR spectrum of **4**. It was supposed that the second position of carbohydrate moiety was occupied by the glucose residue and confirmed by the appearance of the additional signal at δ(C) 61.2 corresponding to C(6) of a glucopyranose residue. Two signals at δ(C) 67.4 and 67.1 corresponding to sulfated hydroxy-methylene carbons of glucopyranose residues were observed in the ^13^C NMR spectrum of **4** indicating the presence of two sulfate groups. The positions of interglycosidic linkages and the consequence of monosaccharides in the chain of **4** were established by analysis of the ROESY and HMBC spectra (Table 6, Appendix A), indicating the presence of linear carbohydrate moiety with the glucose as second unit and sulfated glucose and 3-*O*-methyl-glucose residues as third and terminal monosaccharides, correspondingly. The comparison of the ^13^C NMR spectrum of sugar part of psolusoside G (**4**) with that of earlier known okhotoside B_3,_ isolated from *Cucumaria okhotensis* [19] showed the coincidence of their signals, suggesting the identity of the linear disulfated carbohydrate moieties.

The (−)ESI-MS/MS of **4** demonstrated the fragmentation of [M_2Na_ − Na]^−^ ion at *m/z* 1281.4. The peaks of fragment ions were observed at *m/z*: 1161.5 [M_2Na_ − Na − HSO_4_Na]^−^, 1003.4 [M_2Na_ − Na − HSO_4_Na − C_7_H_12_O_8_SNa (MeGlcSO_3_Na)]^−^, 813.2 [M_2Na_ − Na − C_30_H_43_O_4_ (Agl) − H]^−^, 681.1 [M_2Na_ − Na − C_30_H_43_O_4_ (Agl) − Xyl]^−^, 519.0 [M_2Na_ − Na − C_30_H_43_O_4_ (Agl) − Xyl − Glc]^−^, 255.0 [M_2Na_ − Na − C_30_H_43_O_4_ (Agl) − Xyl − Glc − C_6_H_9_O_8_SNa (GlcSO_3_Na)]^−^, corroborating the aglycone structure and consequence of monosaccharides in psolusoside G (**4**).

All these data indicate that psolusoside G (**4**) is 3β-*O*-[6-*O*-sodium-sulfate-3-*O*-methyl-β-d-glucopyranosyl-(1→3)-6-*O*-sodium-sulfate-β-d-glucopyranosyl-(1→4)-β-d-glucopyranosyl-(1→2)-β-d-xylopyranosyl]-16-ketoholosta-9(11),25-diene.

The sulfation of third or/and fourth monosaccharide residues in the carbohydrate chain when C(4) position of the first xylose residue is not sulfated as in psolusosides E (**2**), F (**3**) and G (**4**) is probably characteristic structural feature of the glycosides of *Psolus fabricii*. It was also observed in a disulfated tetraoside psolusoside A, with sulfate groups at C-6 of the third (glucose) and fourth (3-*O*-methyl-glucose) residues. Monosulfated colochiroside D, isolated first from the sea cucumber *Colochirus robustus* [16] and later from *Psolus fabricii* as well as the disulfated okhotoside B_3_ from *Cucumaria okhotensis* [19], are the glycosides found in the sea cucumbers belonging to other genera, sharing the same structural peculiarity. However, the majority of known sulfated glycosides contain a sulfate group at C-4 of the first xylose residue.

The ^1^H and ^13^C NMR spectra of carbohydrate parts of psolusosides H (**5**) and H_1_ (**6**) were coincident to each other indicating the identity of carbohydrate chains of these glycosides. The presence of three characteristic doublets at δ(H) 4.73 (*J* = 7.5 Hz), 5.19 (*J* = 6.8 Hz), and 4.88 (*J* = 7.9 Hz) in the ^1^H NMR spectra of the carbohydrate chains of **5**, **6** correlated with the HSQC spectra with the signals of anomeric carbons at δ(C) 104.9, 104.4, and 104.5, correspondingly, were indicative of a trisaccharide chain and β-configurations of glycosidic bonds (Table 7, Appendix A). The ^1^H,^1^H-COSY, and 1D TOCSY spectra of **5** and **6** showed the signals of three isolated spin systems assigned to two glucose and one xylose residues (Appendix A). The positions of interglycosidic linkages established by the ROESY and HMBC spectra of **5** and **6** (Table 7, Appendix A) demonstrated cross-peaks between H(1) of the xylose and H(3) (C(3)) of an aglycone, H(1) of the glucose and H(2), (C(2)) of the xylose and H(1) of the terminal unit (glucose), and H(4) (C(4)) of the second unit (glucose). The terminal glucose unit was sulfated by C(6), which was deduced from character signal at δ(C) 67.2 in comparison with the analogous signal of C(6) of the glucose in the second position of the carbohydrate chain, which was observed at δ(C) 61.6. Therefore, the carbohydrate chain of psolusosides H (**5**) and H_1_ (**6**) differed from that of psolusoside G (**4**) in the loss of terminal 3-O-methyl-glucose residue. Actually, the signals in their ^13^C NMR spectra assigning to xylose and glucose (the second unit) residues were coincident. The signal of C(3) of terminal glucose in the spectra of **5**, **6** was shifted up-field to δ(C) 76.9 due to the absence of the glycosylation effect that was observed in the spectrum of **4** (δ(C) 86.4 C(3) of terminal glucose). The carbohydrate chain of psolusosides H (**5**) and H_1_ (**6**) has never been found earlier in the glycosides from sea cucumbers.

The molecular formula of psolusoside H (**5**) was determined to be C_47_H_71_O_21_SNa from the [M_Na_−Na]**^−^** ion peak at *m/z* 1003.4213 (calc. 1003.4214) in the (−)HR-ESI-MS. The ^1^H and ^13^C NMR spectra of aglycone part of psolusoside H (**5**) demonstrated the signals characteristic of the holostane-type aglycone (the signals of 18(20)-lactone at δ(C) 179.0 (C(18)) and 83.6 (C(20))) with 16-keto-group (the signals of C(16) at δ(C) 213.8, C(15) at δ(C) 51.8, and C(17) at δ(C) 63.3 with corresponding proton signals at δ(H) 2.65 (d, *J* = 16.0 Hz, H(15)), and 2.32 (d, *J* = 16.0 Hz, H(15)), as well as 2.87 (s, H(17)) (Table 8, Appendix A). The characteristic signals at δ(C) 121.7 (C(7)), 143.9 (C(8)), and at δ(H) 5.63 (m, H(7)) in the ^13^C and ^1^H NMR spectra of **5** were assigned to 7(8)-double bond in the polycyclic system. The availability of terminal double bond in the side chain of **5** was deduced from the signals at δ(C) 145.4 (C(25)) and 110.3 (C(26)) observed in the ^13^C NMR and two broad singlets at δ(H) 4.70 and 4.69 (H_2_-26) in the ^1^H NMR spectra of psolusoside H (**5**). So, the aglycone of psolusoside H (**5**) is a positional isomer (by the double bond position in polycyclic nucleus) of the aglycone comprising psolusosides E (**2**), F (**3**), and G (**4**). This aglycone was found earlier in the glycosides of sea cucumbers belonging to different orders: *Cucumaria japonica* [20,21], *Pseudocolochirus violaceus* [22] (Cucumariidae, Dendrochirotida), and *Australostichopus mollis* [23] (Stichopodidae, Synallactida) [24].

The (−)ESI-MS/MS of **5** demonstrated the fragmentation of [M_Na_ − Na]^−^ ion at *m/z* 1003.4. The peaks of fragment ions were observed at *m/z*: 535.1 [M_Na_ − Na − C_30_H_43_O_4_ (Agl) − H]^−^, 403.1 [M_Na_ − Na − C_30_H_43_O_4_ (Agl) − Xyl]^−^, 241.0 [M_Na_ − Na − C_30_H_43_O_4_ (Agl) − Xyl − Glc]^−^ corroborating the structure of psolusoside H (**5**).

All these data indicate that psolusoside H (**5**) is 3β-*O*-[6-*O*-sodium-sulfate-β-d-glucopyranosyl-(1→4)-β-d-glucopyranosyl-(1→2)-β-d-xylopyranosyl]-16-ketoholosta-7,25-diene.

The molecular formula of psolusoside H_1_ (**6**) was determined to be C_47_H_73_O_20_SNa from the [M_Na_ − Na]**^−^** ion peak at *m/z* 989.4432 (calc. 989.4421) in the (−)HR-ESI-MS and [M_Na_ + Na]^+^ at *m/z* 1035.4205 (calc. 1035.4206) in the (+)HR-ESI-MS. In the ^1^H and ^13^C NMR spectra of the aglycone part of psolusoside H_1_ (**6**), the signals characteristic of the holostane-type aglycone (the signals of 18(20)-lactone at δ(C) 181.0 (C(18)) and 84.7 (C(20))) with 7(8)-double bond in the polycyclic system (the signals at δ(C) 119.8 (C(7)), 146.5 (C(8)) in the ^13^C NMR, and at δ(H) 5.62 (m, H(7)) in the ^1^H NMR) and terminal double bond in the side chain (the signals at δ(C) 145.5 (C(25)) and 110.6 (C(26)) in the ^13^C NMR and two broad singlets at δ(H) 4.78 and 4.74 (H_2_-26) in the ^1^H NMR) were observed (Table 9, Appendix A). The analysis of ^1^H,^1^H-COSY spectrum of **6** showed the protons H_2_(15)/H_2_(16)/H(17) form the isolated spin system (Appendix A). The signals of C(15), C(16), and C(17) in the ^13^C NMR spectrum of **6** were observed at δ(C) 34.1, 24.5, and 52.9, correspondingly, and were shielded when compared with the signals C(15)–C(17) in the spectrum of psolusoside H (**5**) due to the absence of 16-keto-group in psolusoside H_1_ (**6**). So, the aglycone of psolusoside H_1_ (**6**) differed from that of psolusoside H (**5**) only in the lack of 16-keto-group. Such aglycone was earlier found in the glycosides from sea cucumbers of the order Dendrochirotida: *Colochirus robustus* [16] and *Cucumaria japonica* [20].

The (−)ESI-MS/MS of **6** demonstrated the fragmentation of [M_Na_ − Na]^−^ ion at *m/z* 989.4. The peaks of fragment ions analogous to those for **5** were observed at *m/z*: 535.1 [M_Na_ − Na − C_30_H_45_O_3_ (Agl) − H]^−^, 403.1 [M_Na_ − Na − C_30_H_45_O_3_ (Agl) − Xyl]^−^, 241.0 [M_Na_ − Na − C_30_H_45_O_3_ (Agl) − Xyl − Glc]^−^ corroborating the structure of psolusoside H_1_ (**6**).

All these data indicate that psolusoside H_1_ (**6**) is 3β-*O*-[6-*O*-sodium-sulfate-β-d-glucopyranosyl-(1→4)-β-d-glucopyranosyl-(1→2)-β-d-xylopyranosyl]-holosta-7,25-diene.

The molecular formula of psolusoside I (**7**) was determined to be C_54_H_82_O_29_S_2_Na_2_ from the [M_2Na_ − Na]**^−^** ion peak at *m/z* 1281.4267 (calc. 1281.4286) in the (−)HR-ESI-MS and [M_2Na_ + Na]^+^ at *m/z* 1327.4065 (calc. 1327.4071) in the (+)HR-ESI-MS indicating the presence of two sulfate groups. In the ^1^H and ^13^C NMR spectra (Table 10, Appendix A) of the carbohydrate part of **7** four characteristic doublets at δ(H) 4.63–4.82 (*J* = 7.3–8.1 Hz) and corresponding to them four signals of anomeric carbons at δ(C) 103.9–105.6 were indicative of a tetrasaccharide chain and *β*-configurations of glycosidic bonds (Appendix A). The ^1^H,^1^H-COSY and 1D TOCSY spectra of **7** showed the signals of four isolated spin systems assigned to two xylose and two glucose residues (Appendix A). The positions of interglycosidic linkages established by the ROESY and HMBC spectra of **7** (Table 10, Appendix A) indicated the branched architecture of tetrasaccharide chain when the fourth glucose residue is attached to C(4) of the first (xylose) residue.

The second sugar unit in the chain of psolusoside I (**7**) is a xylose connected to the first xylose residue by β-(1→2)-glycosidic bond. This feature is very rare occurred in the holothurians glycoside’s carbohydrate moieties [25]. The third monosaccharide in the chain is a glucose attached to C(4) of the second (xylose) unit, the fourth residue (glucose) is attached to C-4 of the first xylose unit. Both glucose residues are sulfated by C(6) that was deduced from two signals—at δ(C) 67.6 and 67.9 in the ^13^C NMR spectrum of **7**—demonstrating α-shifting effect of a sulfate group. The tetrasaccharide branched disulfated carbohydrate moiety of psolusoside I (**7**) with the xylose as the second unit has never been found among the sea cucumber glycosides.

The aglycone of psolusoside I (**7**) shared some structural features with the aglycones of psolusosides H (**5**) and H_1_ (**6**). In the ^1^H and ^13^C NMR spectra of the aglycone part of **7,** the signals of holostane-type aglycone (C(18) at δ(C) 180.2 and C(20) at δ(C) 85.5) with 7(8)-double bond in the polycyclic system (the signals at δ(C) 120.2 (C(7)), 145.6 (C(8)), and at δ(H) 5.60 (m, H(7)) and terminal double bond in the side chain (the signals at δ(C) 145.4 (C(25)) and 110.8 (C(26)) in the ^13^C NMR and two broad singlets at δ(H) 4.72 and 4.73 (H_2_-26) in the ^1^H NMR spectra) were observed (Table 11, Appendix A). An isolated spin system formed by the protons H_2_(15)/H(16)/H(17) was deduced from the ^1^H,^1^H-COSY spectrum (Appendix A). The signal of H(16) was observed at δ(H) 5.82 (brq, *J* = 8.6 Hz) and the corresponding signal of C(16) at δ(C) 75.2 indicated the presence of β-*O*-acetic group. Actually, the additional signals corresponding to this group were observed in the ^13^C NMR spectrum of **7** at δ(C) 170.7 (carboxyl carbon) and 21.2 (methyl carbon). The holostane aglycone with 7(8)- and 25(26)-double bonds and 16β-acetoxy group frequently occurred in the glycosides of sea cucumbers [2,16,19].

The (−)ESI-MS/MS of **7** demonstrated the fragmentation of [M_2Na_ − Na]^−^ ion at *m/z* 1281.4. The peaks of fragment ions were observed at *m/z*: 769.1 [M_2Na_ − Na − C_32_H_47_O_5_ (Agl) − H]^−^, 505.2 [M_2Na_ − Na − C_32_H_47_O_5_ (Agl) − C_6_H_10_O_8_SNa (GlcSO_3_Na)]^−^, 373.0 2 [M_2Na_ − Na − C_32_H_47_O_5_ (Agl) − GlcSO_3_Na − Xyl]^−^, 241.0 2 [M_2Na_ − Na − C_32_H_47_O_5_ (Agl) − GlcSO_3_Na − 2Xyl]^−^. The (+)ESI-MS/MS of **7** demonstrated the fragmentation of [M_2Na_ + Na]^+^ ion at *m/z* 1327.4. The peaks of fragment ions were observed at *m/z*: 1207.5 [M_2Na_ + Na − NaHSO_4_]^+^, 1147.4 [M_2Na_ + Na − NaHSO_4_ − CH_3_COOH]^+^, 1063.4 [M_2Na_ + Na − C_6_H_10_O_8_SNa (GlcSO_3_Na)]^+^, 1003.4 [M_2Na_ + Na – GlcSO_3_Na – CH_3_COO]^+^, 931.4 [M_2Na_ + Na – GlcSO_3_Na − Xyl + H]^+^, 833.1 [M_2Na_ + Na − C_32_H_47_O_5_ (Agl) +H]^+^. All these data exhaustively confirmed the structure psolusoside I (**7**) deduced by analyses of NMR data.

All these data indicate that psolusoside I (**7**) is 3β-*O*-{6-*O*-sodium sulfate-β-d-glucopyranosyl-(1→4)-β-d-xylopyranosyl-(1→2)-[6-*O*-sodium sulfate-β-d-glucopyranosyl-(1→4)]-β-d-xylopyranosyl}-16β-acetoxyholosta-7,25-diene.

### 2.2. Bioactivity of the Glycosides

The cytotoxic activities of compounds **2**–**7** as well as earlier known psolusoside A (was used as positive control [26]) against mouse erythrocytes (hemolytic activity), the ascite form of mouse Ehrlich carcinoma cells, neuroblastoma Neuro 2A cells, and normal epithelial JB-6 cells are presented in Table 12. The investigated substances, except psolusoside I (**7**) containing a xylose in the second position of carbohydrate chain, demonstrate high hemolytic action, but the majority of them are not active or slightly active against mouse Ehrlich carcinoma cells (ascite form) (except psolusoside A which has a moderate cytotoxic action).

Psolusosides A and E (**2**) are the strongest cytotoxins in this series. The cytotoxicities of psolusosides F (**3**), H (**5**), H_1_ (**6**), and I (**7**) against normal JB-6 cells are comparable with those against neuroblastoma Neuro 2A cells. However, there is one interesting exception: Psolusoside G (**4**) (disulfated linear tetraoside with a glucose as second sugar in the chain) is not cytotoxic against normal JB-6 cells but demonstrates high activity against Neuro 2A cells. It opens a possibility to study this compound on models of neurodegenerative diseases. Noteworthy, the cytotoxicity of psolusosides H (**5**) and H_1_ (**6**) (the glycosides having trisaccharide chains) and psolusoside I (**7**) (the compound with tetrasaccharide branched carbohydrate chain and a xylose as the second unit) are similar to that of linear tetraosides—psolusosides A, E (**2**), F (**3**), and G (**4**)—whereas it is known that the presence of linear tetrasaccharide chain is one of the necessary conditions for the displaying of membranolytic activity [4,6].

The influence of the psolusosides A, B (**1**), E (**2**), F (**3**), H (**5**), H_1_ (**6**), and I (**7**) on cell viability, formation, and growth of colonies of human colorectal adenocarcinoma HT-29 cells was checked. HT-29 cells were treated with various concentrations (0–20 μM) of compounds **1**–**3**, **5**–**7** and earlier known psolusoside A for 24 h, and then cell viability was assessed by the MTS assay. It was shown that all investigated compounds are not cytotoxic against HT-29 cells at the dose of 20 μM. The concentrations 10 μM were chosen for the investigation of the glycosides influence on the colony formation of HT-29 cells in soft agar assay. The data concerning inhibitory activity of psolusosides A, B (**1**), E (**2**), F (**3**), H (**5**), H_1_ (**6**), and I (**7**) on colony formation of HT-29 cells are presented in Table 13. The highest inhibition of colony formation and growth of HT-29 cells demonstrate psolusosides E (**2**) and F (**3**) (ICCF_50_ 0.1 μM and 0.5 μM, respectively) having the holostane aglycones and linear tetrasaccharide monosulfated carbohydrate chains with the quinovose residue as the second sugar unit. The inhibitory action of compounds **6** and **7** is observed only at the doses 9 and 10 μM, respectively. While psolusosides A, B (**1**), and H (**5**) did not inhibit the colony formation and growth of HT-29 cells for 50% under concentration of 10 μM. Thus, the presence of trisaccharide (as in psolusosides H (**5**) and H_1_ (**6**)) or branched tetrasaccharide (as in psolusosides B (**1**) and I (**7**)) chains, even in combination with the holostane aglycones (as in **5**–**7**), cause the loss of this type of bioactivity. 

We also try to study synergic effects of these compounds (0.05 μM) and radioactive irradiation (2 Gy) of HT-29 cells. The number of colonies of HT-29 cells was found to be decreased after radiation exposure at a dose of 2 Gy, but synergic effects of the glycosides and radioactive irradiation (2 Gy) decreasing the number of colonies was not observed.

## 3. Materials and Methods

### 3.1. General Experimental Procedures

Specific rotation, Perkin-Elmer 343 Polarimeter; NMR, Bruker Avance III 500 (Bruker BioSpin GmbH, Rheinstetten, Germany) (500.13/125.77 MHz) or Avance III 700 Bruker FT-NMR (Bruker BioSpin GmbH, Rheinstetten, Germany) (700.13/176.04 MHz) (^1^H/^13^C) spectrometers; ESI MS (positive and negative ion modes), Agilent 6510 Q-TOF (Agilent Technologies, Santa Clara, CA, USA) apparatus, sample concentration 0.01 mg/mL; HPLC, Agilent 1100 apparatus with a differential refractometer; columns Supelco Ascentis RP-Amide (10×250 mm, 5 μm), Kromasil Cellucoat RP (4.6×150 mm, 5 μm), and Supelcosil LC-Si (4.6×150 mm, 5 μm).

### 3.2. Animal Material

Specimens of the sea cucumber *Psolus fabricii* (family Psolidae; order Dendrochirotida) were collected in the Sea of Okhotsk near Onekotan Island (Kurile Islands). Sampling was performed with a scallop dredge in August to September,1982 at a depth of 100 m during expedition works on fishing seiners “Mekhanik Zhukov” and “Dalarik”. Sea cucumbers were identified by Prof. V.S. Levin; voucher specimens are preserved in A.V. Zhirmunsky National Scientific Center of Marine Biology, Vladivostok, Russia.

### 3.3. Extraction and Isolation

The sea cucumbers were minced and extracted twice with refluxing 60% EtOH. The extract was evaporated to water residuum and lyophilized followed by the extraction with CHCl_3_/MeOH (1:1). The obtained extract was evaporated and submitted to the subsequent extraction by EtOAc/H_2_O to remove the lipid fraction. The water layer remaining after this extraction was chromatographed on a Polychrom-1 column (powdered Teflon, Biolar, Latvia). The glycosides were eluted with 50% EtOH, evaporated, and subsequently chromatographed on Si gel columns with CHCl_3_/EtOH/H_2_O (100:75:10), (100:100:17), (100:125:25) as mobile phase to give subfractions III–VIII containing different groups of glycosides. These subfractions kept at the temperature −18 °C, then were submitted to HPLC on silica-based (Supelcosil LC-Si) and reversed phase (Supelco Ascentis RP-Amide, Kromasil Cellucoat RP) columns with different solvent systems as mobile phase. The subfraction III was chromatographed on Supelco Ascentis RP-Amide column with 35% acetonitrile (CH_3_CN) and then on Supelcosil LC-Si with CHCl_3_/MeOH/H_2_O (85/20/2) as mobile phases to give 10 mg of psolusoside E (**2**). The subfraction IV was submitted to HPLC on Supelco Ascentis RP-Amide column with CH_3_CN/H_2_O/NH_4_OAc (1 M water solution) (45/54/1) to give 1.4 mg of psolusoside H_1_ (**6**) as well as five other fractions. Fraction 3 was chromatographed on the column Kromasil Cellucoat-RP with 23% CH_3_CN that resulted in isolation of colochiroside D (2.5 mg). The subsequent HPLC of the fraction 4 on Supelco Ascentis RP-Amide column with CH_3_CN/H_2_O/NH_4_OAc (40/59/1) gave 1.4 of psolusoside H (**5**). Fraction 5 was submitted to HPLC on Supelcosil LC-Si column with CHCl_3_/MeOH/H_2_O (65/20/2) as mobile phase to give pure psolusoside F (**3**) (1.4 mg). The subfraction V obtained after Si-gel column chromatography was submitted to HPLC on Supelco Ascentis RP-Amide column with CH_3_CN/H_2_O/NH_4_OAc (40/59/1) resulted in isolation of known psolusoside A (37 mg). HPLC of subfraction VI on the same column with other ratio of the solvents used before—(43/55/2)—gave 46.5 mg of psolusoside G (**4**). Subfraction VII was submitted to HPLC on Kromasil Cellucoat-RP column with 14% CH_3_CN followed by HPLC on Supelcosil LC-Si column with CHCl_3_/MeOH/H_2_O (65/20/2) as mobile phase to give 1.1 mg of psolusoside I (**7**). Psolusoside B (**1**) (67 mg) was isolated from subfraction VIII as result of HPLC on Supelco Ascentis RP-Amide column with CH_3_CN/H_2_O/NH_4_OAc (35/64/1) as mobile phase.

#### 3.3.1. Psolusoside B (1)

Colorless powder; [α]${}_{D}^{20}$ –60 (*c* 0.1, 50% MeOH). NMR: See Table 1 and Table 2. (+)HR-ESI-MS *m/z*: 1357.4169 (calc. 1357.4176) [M_2Na_ + Na]^+^, 690.2039 (calc. 690.2034) [M_2Na_ + 2Na]^2+^; (+)ESI-MS/MS *m/z*: 1297.4 [M_2Na_ + Na − CH_3_COOH]^+^, 1237.4 [M_2Na_ + Na − NaHSO_4_]^+^, 1177.4 [M_2Na_ + Na − CH_3_COOH − NaHSO_4_]^+^, 1117.4 [M_2Na_ + Na − 2NaHSO_4_]^+^, 1075.5 [M_2Na_ + Na − C_6_H_10_O_9_SNa (GlcSO_3_Na)]^+^, 913.4 [M_2Na_ + Na − GlcSO_3_Na − Glc]^+^, 863.1 [M_2Na_ + Na − C_32_H_47_O_4_ (Agl) + H]^+^, 743.1 [M_2Na_ + Na − C_32_H_47_O_4_ (Agl) − NaHSO_4_]^+^, 581.1 [M_2Na_ + Na − C_32_H_47_O_4_ (Agl) − C_6_H_10_O_9_SNa (GlcSO_3_Na)]^+^, 449.0 [M_2Na_ + Na − C_32_H_47_O_4_ (Agl) − C_6_H_10_O_9_SNa (GlcSO_3_Na) − Xyl]^+^, 287.0 [M_2Na_ + Na − C_32_H_47_O_4_ (Agl) − C_6_H_10_O_9_SNa (GlcSO_3_Na) − Xyl − Glc]^+^.

#### 3.3.2. Psolusoside E (2)

Colorless powder; [α]${}_{D}^{20}$ −25 (*c* 0.1, 50% MeOH). NMR: See Table 3 and Table 4. (−)HR-ESI-MS *m/z*: 1163.4945 (calc. 1163.4950) [M_Na_ − Na]^−^; (−)ESI-MS/MS *m/z*: 1163.5 [M_Na_ − Na]^−^, 987.4 [M_Na_ − Na − MeGlc + H]^−^, 695.2 [M_Na_ − Na − C_30_H_43_O_4_ (Agl) − H]^−^, 563.1 [M_Na_ − Na − C_30_H_43_O_4_ (Agl) − Xyl]^−^, 417.1 [M_Na_ − Na − C_30_H_43_O_4_ (Agl) − Xyl − Qui]^−^, 241.0 [M_Na_ − Na − C_30_H_43_O_4_ (Agl) − Xyl − Qui − MeGlc]^−^.

#### 3.3.3. Psolusoside F (3)

Colorless powder; [α]${}_{D}^{20}$ −50 (*c* 0.1, 50% MeOH). NMR: See Table 3 and Table 5. (−)HR-ESI-MS *m/z*: 1163.4952 (calc. 1163.4950) [M_Na_ − Na]^−^; (−)ESI-MS/MS *m/z*: 1163.5 [M_Na_ − Na]^−^, 695.2 [M_Na_ − Na − C_30_H_43_O_4_ (Agl) − H]^−^, 563.1 [M_Na_ − Na − C_30_H_43_O_4_ (Agl) − Xyl]^−^, 417.1 [M_Na_ − Na − C_30_H_43_O_4_ (Agl) − Xyl − Qui]^−^, 255.0 [M_Na_ − Na − C_30_H_43_O_4_ (Agl) − Xyl − Qui − Glc]^−^.

#### 3.3.4. Psolusoside G (4)

Colorless powder; [α]${}_{D}^{20}$ −49 (*c* 0.1, 50% MeOH). NMR: See Table 3 and Table 6. (−)HR-ESI-MS *m/z*: 1281.4313 (calc. 1281.4286) [M_2Na_ − Na]^−^; (−)ESI-MS/MS *m/z*: 1281.4 [M_2Na_ − Na]^−^, 1161.5 [M_2Na_ − Na − HSO_4_Na]^−^, 1003.4 [M_2Na_ − Na − HSO_4_Na − C_7_H_12_O_8_SNa (MeGlcSO_3_Na)]^−^, 813.2 [M_2Na_ − Na − C_30_H_43_O_4_ (Agl) − H]^−^, 681.1 [M_2Na_ − Na − C_30_H_43_O_4_ (Agl) − Xyl]^−^, 519.0 [M_2Na_ −Na − C_30_H_43_O_4_ (Agl) − Xyl − Glc]^−^, 255.0 [M_2Na_ − Na − C_30_H_43_O_4_ (Agl) − Xyl − Glc − C_6_H_9_O_8_SNa (GlcSO_3_Na)]^−^.

#### 3.3.5. Psolusoside H (5)

Colorless powder; [α]${}_{D}^{20}$ −35 (*c* 0.1, 50% MeOH). NMR: See Table 7 and Table 8. (−)HR-ESI-MS *m/z*: 1003.4213 (calc. 12003.4214) [M_Na_ − Na]^−^; (−)ESI-MS/MS *m/z*: 1003.4 [M_Na_ − Na]^−^, 535.1 [M_Na_ − Na − C_30_H_43_O_4_ (Agl) − H]^−^, 403.1 [M_Na_ − Na − C_30_H_43_O_4_ (Agl) − Xyl]^−^, 241.0 [M_Na_ − Na − C_30_H_43_O_4_ (Agl) − Xyl − Glc]^−^.

#### 3.3.6. Psolusoside H_1_ (6)

Colorless powder; [α]${}_{D}^{20}$ −23 (*c* 0.1, 50% MeOH). NMR: See Table 7 and Table 9. (−)HR-ESI-MS *m/z*: 989.4432 (calc. 989.4421) [M_Na_ − Na]^−^; (+)HR-ESI-MS *m/z*: 1035.4205 (calc. 1035.4206) [M_Na_ + Na]^+^; (−)ESI-MS/MS *m/z*: 989.4 [M_Na_ − Na]^−^, 535.1 [M_Na_ − Na − C_30_H_45_O_3_ (Agl) − H]^−^, 403.1 [M_Na_ − Na − C_30_H_45_O_3_ (Agl) − Xyl]^−^, 241.0 [M_Na_ − Na − C_30_H_45_O_3_ (Agl) − Xyl − Glc]^−^.

#### 3.3.7. Psolusoside I (7)

Colorless powder; [α]${}_{D}^{20}$ −17 (*c* 0.1, 50% MeOH). NMR: See Table 10 and Table 11. (−)HR-ESI-MS *m/z*: 1281.4267 (calc. 1281.4286) [M_2Na_ − Na]^−^; (+)HR-ESI-MS *m/z*: 1327.4065 (calc. 1327.4071) [M_2Na_ + Na]^+^; (−)ESI-MS/MS *m/z*: 1281.4 [M_2Na_ − Na]^−^, 769.1 [M_2Na_ − Na − C_32_H_47_O_5_ (Agl) − H]^−^, 505.2 [M_2Na_ − Na – C_32_H_47_O_5_ (Agl) − C_6_H_10_O_8_SNa (GlcSO_3_Na)]^−^, 373.0 2 [M_2Na_ − Na − C_32_H_47_O_5_ (Agl) − GlcSO_3_Na − Xyl]^−^, 241.0 2 [M_2Na_ − Na − C_32_H_47_O_5_ (Agl) − GlcSO_3_Na − 2Xyl]^−^; (+)ESI-MS/MS *m/z*: 1327.4 [M_2Na_ + Na]^+^, 1207.5 [M_2Na_ + Na −NaHSO_4_]^+^, 1147.4 [M_2Na_ + Na − NaHSO_4_ − CH_3_COOH]^+^, 1063.4 [M_2Na_ + Na − C_6_H_10_O_8_SNa (GlcSO_3_Na)]^+^, 1003.4 [M_2Na_ + Na − GlcSO_3_Na − CH_3_COO]^+^, 931.4 [M_2Na_ + Na − GlcSO_3_Na − Xyl + H]^+^, 833.1 [M_2Na_ + Na − C_32_H_47_O_5_ (Agl) +H]^+^.

### 3.4. Cell Culture

The museum tetraploid strain of murine ascite Ehrlich carcinoma cells from the All-Russian Oncology Center (Moscow, Russia) was used. The cells were separated from the ascites, which were collected on day 7 after inoculation in mouse CD-1 line. The cells were washed of the ascites triply and resuspended in RPMI-1640 medium containing 8 μg/mL gentamicin (BioloT, St-Petersburg, Russia). Neuroblastoma Neuro 2A cells were cultured in DMEM medium containing 10% fetal bovine serum (FBS), (BioloT, St-Petersburg, Russia) and normal epithelial JB-6 cells were cultured in DMEM medium containing 5% fetal bovine serum (FBS), (BioloT, St-Petersburg, Russia) and 1% penicillin/streptomycine (Invitrogen). The HT-29 (ATCC # HTB-38) human colon cancer cell line was grown in monolayer in McCoy’s 5A medium supplemented with 10% (v/v) heat-inactivated FBS, 2 mM l-glutamine, and 1% penicillin-streptomycin in a humidified atmosphere containing 5% CO_2_. Cells were maintained in a sterile environment and kept in an incubator at 5% CO_2_ and 37 °C to promote growth. HT-29 cells were sub-cultured every 3–4 days by rinsing with phosphate buffered saline (PBS), adding trypsin to detach the cells from the tissue culture flask, and transferring 10%–20% of the harvested cells to a new flask containing fresh growth media.

### 3.5. Cytotoxic Activity

#### 3.5.1. Nonspecific Esterase Activity Assay

Cytotoxic activity against ascite form of mouse Ehrlich carcinoma cells was investigated by nonspecific esterase activity assay. In total, 10µL of the test substance solution and 100 μL of the cell suspension were placed into each well of a 96-well microplate. The plate was incubated in a CO_2_ incubator at 37 °C for 1 or 24 h. A stock solution of the probe fluorescein diacetate (FDA; Sigma, St. Louis, MO, USA) in DMSO (1 mg/mL) was prepared. After incubation of the cells with test compounds, 10 μL of FDA solution (50 μg/mL) was added to each well and the plate was incubated at 37 °C for 15 min. Cells were washed with PBS, and fluorescence was measured with a Fluoroskan Ascent plate reader (Thermo Labsystems, Helsinky, Finland) at λex = 485 nm and λem = 518 nm. All experiments were repeated in triplicate. Cytotoxic activity was expressed as the percent of cell viability. Nonspecific esterase activity assay has been used for determination of cytotoxicity against neuroblastoma Neuro 2A and normal epithelial JB-6 cells.

#### 3.5.2. MTT Assay

The solutions of tested substances in different concentrations (20 µL) and cell suspension (200 µL) were added in wells of 96-well plate and incubated over night at 37 °C and 5% CO_2_. After incubation, the cells were sedimented by centrifugation, 200 µL of medium from each well were collected, and 100 µL of pure medium were added. Then, 10 µL of MTT solution 5 µg/mL (Sigma, St. Louis, MO, USA) were added in each well. The plate was incubated for 4 h, after which 100 µL SDS-HCl were added to each well, and the plate was incubated at 37 °C for 4–18 h. Optical density was measured at 570 nm and 630–690 nm. Cytotoxic activity of the substances was calculated as the concentration that caused 50% metabolic cell activity inhibition (IC_50_). The MTT assay has been used for determination of cytotoxic activity against Ehrlich carcinoma cells.

#### 3.5.3. MTS Assay

The HT-29 cells (1.0 × 10^4^/well) were seeded in 96-well plates for 24 h at 37 °C in 5% CO_2_ incubator. The cells were treated with the tested substances at concentrations range from 0 to 20 μM for an additional 24 h. Subsequently, cells were incubated with 15 μL MTS reagent for 3 h and the absorbance in each well was measured at 490/630 nm using microplate reader “Power Wave XS” (Bio Tek, Winooski, VT, USA). All the experiments were repeated three times, and the mean absorbance values were calculated. The results are expressed as the percentage of inhibition that produced a reduction in absorbance by compound’s treatment compared to the non-treated cells (control).

### 3.6. Hemolytic Activity

Blood was taken from CD-1 mice (18–20 g). The mice were anesthetized with diethyl ether, their chests were rapidly opened, and blood was collected in cold (4 °C) 10 mM phosphate-buffered saline, pH 7.4 (PBS) without anticoagulant. Erythrocytes were washed 3 times in PBS using at least 10 vol. of washing solution by centrifugation (2000 rpm) for 5 min. Erythrocytes were used at a concentration that provided an optical density of 1.0 at 700 nm for a non-hemolyzed sample. Then, 20 μL of a water solution of test substance with a fixed concentration was added to a well of a 96-well plate containing 180 μL of the erythrocyte suspension. Erythrocyte suspension was incubated with substances for 24 h at 37 °C. After that, the optical density of the obtained solutions was measured and ED_50_ for hemolytic activity of each compound was calculated.

### 3.7. Soft Agar Assay

The HT-29 cells (8.0 × 10^3^) were seeded in 6-well plate and treated with the tested compounds at concentrations range 0–10 μM in 1 mL of 0.3% Basal Medium Eagle (BME) agar containing 10% FBS, 2mM L-glutamine, and 25 μg/mL gentamicin. The cultures were maintained at 37 °C in a 5% CO_2_ incubator for 14 days, and the cell’s colonies were scored using a microscope “Motic AE 20” (Scientific Instrument Company, Campbell, CA, USA) and the Motic Image Plus version 2.0 (Scientific Instrument Company, Campbell, CA, USA) computer program.

### 3.8. Radiation Exposure

Irradiation was delivered at room temperature using single doses of X-ray system XPERT 80 (KUB Technologies, Inc, Milford, CT, USA). The dose 2 Gy was used for colony formation assay. The absorber dose was measured using X-ray radiation clinical dosimeter DRK-1 (Moscow, Russia).

### 3.9. Cell Irradiation

The HT-29 cells (6.0 × 10^5^) were plated at 60 mm dishes and incubated for 24 h. After the incubation, the cells were cultured in the presence or absence of 0.05 μM of tested compounds for additional 24 h before irradiation at the dose of 2 Gy. Immediately after irradiation, the cells were returned to the incubator for recovery. Three hours later, the cells were harvested and used for soft agar assay to establish the synergism of radioactive irradiation and investigated compounds effects on colony formation of tested cells.

## 4. Conclusions

Seven individual compounds **1**–**7,** including mono- and disulfated, linear, and branched tetraosides, as well as monosulfated triosides, were isolated from the sea cucumber *Psolus fabricii* in addition to recently obtained non-sulfated hexaosides [14,15]. The structural analysis of the glycosides **1**–**7** demonstrated the variability of their aglycones and carbohydrate chains. Five aglycones and six different carbohydrate chains were found in these compounds. Although all the aglycones were earlier known, five types of sugar chains in these glycosides were new. Three linear tetraosides—psolusosides E (**2**), F (**3**), and G (**4**)—are biogenetically interrelated. These compounds share the same aglycone and differ from each other in positions and numbers of sulfate groups and in the nature of the second monosaccharide residue in the carbohydrate chain (quinovose or glucose). The compounds **2**–**4** and two disulfated branched tetraosides —psolusosides B (**1**) and I (**7**)—altogether are a good illustration of the mosaicism of carbohydrate chain biosynthesis. Firstly, diverse monosaccharide residues (quinovose, glucose, or xylose) can glycosylate C(2) of the first xylose unit; secondly, the fourth (terminal) monosaccharide residue can bind to C(3) of the third monosaccharide unit (glucose), which resulted in the formation of linear chains of psolusosides E (**2**), F (**3**), and G (**4**), or to C(4) of the first (xylose) residue resulted in the formation of branched chains of the glycosides **1** and **7**. At that, carbohydrate chain of psolusosides H (**5**) and H_1_ (**6**) can be biosynthetic precursor for the both type of tetrasaccharide chains—linear chain of psolusoside G (**4**) and branched chain of psolusoside B (**1**) (Figure 2).

The aglycones of triosides, psolusosides H (**5**) and H_1_ (**6**), and branched tetraosides, psolusosides B (**1**) and I (**7**), are structurally diverse. These aglycones belong either to non-holostane (with 18(16)-lactone—as in **1**) or to holostane (with 18(20)-lactone as in **5**–**7**) types and share the presence of 7(8)- and 25(26)-double bonds. It is known that holostane-type aglycones are biosynthesized via the hydroxylation of C(20) in triterpene precursor followed by C(18) oxidation, resulting in the formation of 18(20)-lactone. When the hydroxyl groups are simultaneously present at C(16) and C(20) of 18-carboxylated derivative, the formation of 18(16)-lactone occurred [25]. This situation is obviously realized in the aglycone of psolusoside B (**1**). The acetylation of С(16) (as in psolusoside I (**7**)) or oxidation of the corresponding hydroxyl group at C(16) to a carbonyl (as in psolusoside H (**5**)) prevent the formation of 18(16)-lactone and lead to the synthesis of holostane derivatives. However, the incorporation of this type of functionalities to C(16) could be also realized after the 18(20)-lactonization due to the mosaic type of biosynthesis of triterpene glycosides of sea cucumbers. Hence, unsubstituted at C(16) precursors of holostane aglycone as well as their acetates and 16-keto derivatives may give holostane glycosides after previous 18(20)-lactonization, as it realized at biosynthesis of **2**–**4** and **5**–**7** (Figure 3).

## Figures and Tables

**Figure 1 marinedrugs-17-00358-f001:**
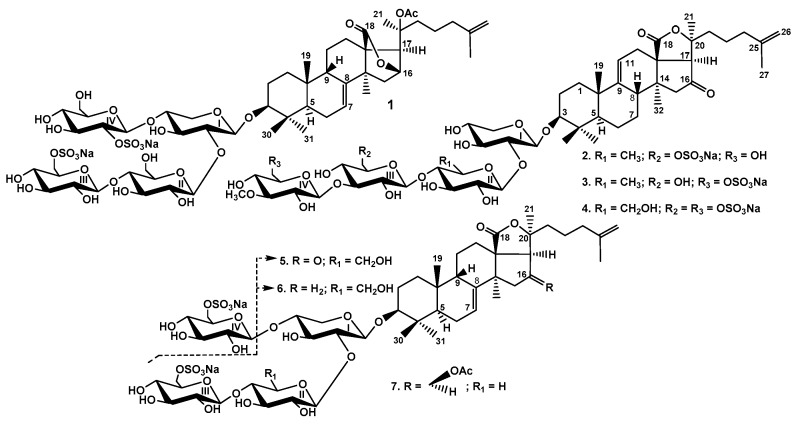
Chemical structure of the glycosides isolated from *Psolus fabricii:*
**1**—psolusoside B; **2**—psolusoside E; **3**—psolusoside F; **4**—psolusoside G; **5**—psolusoside H; **6**—psolusoside H_1_; **7**—psolusoside I.

**Figure 2 marinedrugs-17-00358-f002:**
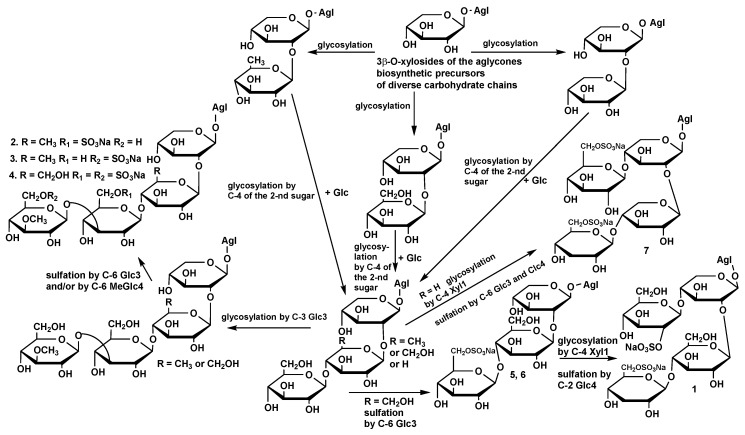
The hypothetic scheme of the carbohydrate chains biosynthesis of the glycosides of *P. fabricii*.

**Figure 3 marinedrugs-17-00358-f003:**
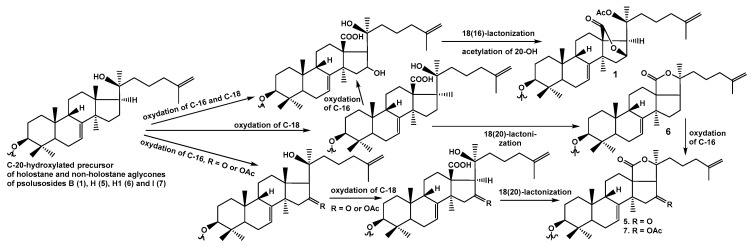
The hypothetic scheme of biosynthesis of holostane and non-holostane aglycones of the glycosides of *P. fabricii*.

**Table 1 marinedrugs-17-00358-t001:** ^13^C and ^1^H NMR chemical shifts and HMBC and ROESY correlations of aglycone moiety of psolusoside B (**1**). ^a^ Recorded at 176.04 MHz in C_5_D_5_N/D_2_O (4/1). ^b^ Recorded at 700.13 MHz in C_5_D_5_N/D_2_O (4/1).

Position	δ_C_ mult. ^a^	δ_H_ mult. ^b^ (*J* in Hz)	HMBC	ROESY
1	35.6 CH_2_	1.41 m		
		1.36 m		
2	26.7 CH_2_	1.96 m		
		1.78 m		H-19, H-30
3	89.3 CH	3.14 (dd, 3.9; 11.8)	C: 4, 30, 31, C-1 Xyl1	H-1, H-5, H-31, H-1 Xyl1
4	39.2 C			
5	47.6 CH	0.84 (dd, 3.8; 11.8)	C: 4, 10, 19, 30, 31	H-3, H-31
6	23.1 CH_2_	1.87 m	C: 5, 10	H-31
		1.75 m		
7	122.8 CH	5.56 (brd, 6.8)	C: 6, 9	H-15, H-32
8	147.0 C			
9	45.9 CH	2.97 (brd, 13.9)		H-19
10	35.4 C			
11	21.9 CH_2_	1.99 m		
		1.47 m		
12	20.0 CH_2_	2.33 (d, 12.9)	C: 13, 14, 18	
		2.02 m		
13	54.9 C			
14	45.6 C			
15	44.2 CH_2_	2.10 m	C: 8, 16, 17	H-7
		2.07 m	C: 14, 32	
16	79.7 CH	4.93 brs	C: 13, 14, 18	H-21, H-22, H-23
17	60.5 CH	3.05 s	C: 13, 14, 18, 20, 21, 22	H-15, H-21, H-22, H-23
18	182.3 C			
19	23.8 CH_3_	0.88 s	C: 1, 5, 9, 10	H-2, H-6, H-9, H-30
20	84.1 C			
21	23.6 CH_3_	1.62 s	C: 17, 20, 22	H-16, H-17
22	37.7 CH_2_	2.23 (dt, 4.5; 13.2)		
		1.82 m	C: 17, 21, 23	H-16
23	21.7 CH_2_	1.47 m	C: 22, 24, 25	
24	37.7 CH_2_	1.97 (dd, 6.9; 13.1)	C: 22, 23, 25, 26	H-26
25	145.4 C			
26	110.7 CH_2_	4.73 brs	C: 24, 25, 27	
27	22.1 CH_3_	1.65 s	C: 24, 25, 26	H-26
30	17.1 CH_3_	0.98 s	C: 3, 4, 5, 31	H-2, H-6, H-19
31	28.5 CH_3_	1.12 s	C: 3, 4, 5, 30	H-3, H-5, H-6, H-1 Xyl1
32	34.2 CH_3_	1.39 s	C: 8, 13, 14, 15	H-7, H-15, H-17
OAc	170.9 C			
	21.6 CH_3_	2.06 s	OAc	

**Table 2 marinedrugs-17-00358-t002:** ^13^C and ^1^H NMR chemical shifts and HMBC and ROESY correlations of carbohydrate moiety of psolusoside B (**1**). ^a^ Recorded at 176.04 MHz in C_5_D_5_N/D_2_O (4/1). ^b^ Bold is interglycosidic positions. ^c^ Italic is sulphate position. ^d^ Recorded at 700.13 MHz in C_5_D_5_N/D_2_O (4/1). Multiplicity by 1D TOCSY.

Atom	δ_C_ mult. ^a,b,c^	δ_H_ mult. ^d^ (*J* in Hz)	HMBC	ROESY
Xyl1 (1→C-3)				
1	104.8 CH	4.56 (d, 7.3)	C: 3; C: 5 Xyl1	H-3; H-3, 5 Xyl1
2	**81.0** CH	4.01 (t, 8.0)	C: 1 Glc2; C: 1, 3 Xyl1	H-1 Glc2
3	75.2 CH	4.20 (t, 8.8)	C: 2, 4 Xyl1	H-1 Xyl1
4	**78.6** CH	4.08 (dt, 5.6; 9.6)	C: 1 Clc4; C: 5 Xyl1	H-1 Glc4
5	63.6 CH_2_	4.43 (dd, 5.2; 12.1)	C: 1, 3, 4 Xyl1	
		3.73 (brt, 11.3)	C: 1 Xyl1	H-1 Xyl1
Glc2 (1→2Xyl1)				
1	104.1 CH	5.11 (d, 7.8)	C: 2 Xyl1; C: 5 Glc2	H-2 Xyl1; H-3, 5 Glc2
2	75.1 CH	3.82 (t, 7.8)	C: 1, 3 Glc2	
3	75.2 CH	3.96 (t, 8.7)	C: 2, 4 Glc2	H-1 Glc2
4	**82.2** CH	3.87 (t, 8.7)	C: 1 Glc3; C: 5, 6 Glc2	H-1 Glc3
5	75.9 CH	3.70 (dt, 2.9; 9.7)		H-1, 3 Glc2
6	61.4 CH_2_	4.30 (dd, 2.9; 12.3)		
		4.25 (dd, 4.6; 12.2)	C: 4, 5 Glc2	
Glc3 (1→4Glc2)				
1	104.5 CH	4.81 (d, 7.9)	C: 4 Glc2	H-4 Glc2; H-5 Glc3
2	74.1 CH	3.79 (t, 9.2)	C: 1, 3, 4 Glc3	H-4 Glc3
3	**76.8** CH	4.07 (t, 9.2)	C: 2, 4 Glc3	H-1 Glc3
4	70.7 CH	3.90 (t, 9.2)	C: 3, 5, 6 Glc3	
5	75.5 CH	4.03 (dd, 4.6; 10.1)		H-1 Glc3
6	*67.5* CH_2_	5.01 (d, 10.1)	C: 4 Glc3	
		4.64 (dd, 6.7; 11.1)	C: 5 Glc3	
Glc4 (1→4Xyl1)				
1	100.9 CH	4.92 (d, 7.8)	C: 4 Xyl1	H-4 Xyl1; H-3, 5 Glc4
2	*80.6* CH	4.74 (t, 8.9)	C: 1, 3 Glc4	
3	76.8 CH	4.28 (t, 8.9)	C: 2, 4 Glc4	H-1, 5 Glc4
4	70.7 CH	3.90 (t, 8.9)	C: 3, 5, 6 Glc4	
5	77.4 CH	3.84 (dd, 4.6; 10.2)	C: 4 Glc4	H-1 Glc4
6	61.8 CH_2_	4.32 (dd, 2.5; 12.1)	C: 4 Glc4	
		4.01 (dd, 6.4; 12.1)	C: 4, 5 Glc4	

**Table 3 marinedrugs-17-00358-t003:** ^13^C and ^1^H NMR chemical shifts and HMBC and ROESY correlations of aglycone moiety of psolusosides E (**2**), F (**3**), G (**4**). ^a^ Recorded at 176.04 MHz in C_5_D_5_N/D_2_O (4/1). ^b^ Recorded at 700.13 MHz in C_5_D_5_N/D_2_O (4/1).

Position	δ_C_ mult. ^a^	δ_H_ mult. ^b^ (*J* in Hz)	HMBC	ROESY
1	36.2 CH_2_	1.89 m		H-11, H-19
		1.52 m		H-3, H-5, H-11
2	27.0 CH_2_	2.30 m		
		2.02 m		H-19, H-30
3	88.7 CH	3.31 (dd, 4.8; 11.6)	C: 4, 30, 31, C-1 Xyl1	H-1, H-5, H-31, H-1 Xyl1
4	39.6 C			
5	52.8 CH	0.99 (brd, 12.0)	C: 4, 10, 19, 30	H-1, H-3, H-7, H-31
6	21.0 CH_2_	1.75 m		
		1.57 m		H-19, H-30
7	28.4 CH_2_	1.62 m		H-15
		1.27 m		H-5, H-32
8	38.6 CH	3.29 m	C: 9	H-15, H-19
9	151.2 C			
10	39.8 C			
11	110.9 CH	5.35 m	C: 8, 13	H-1
12	32.0 CH_2_	2.48 m	C: 14	H-21
		2.52 m	C: 9, 11, 13, 14, 18	H-17, H-32
13	55.6 C			
14	41.9 C			
15	51.8 CH_2_	2.39 d (15.6)	C: 13, 16, 17, 32	H-7, H-32
		2.23 d (15.6)	C: 14, 16, 32	H-8
16	212.9 C			
17	61.2 CH	2.80 s	C: 12, 13, 16, 18, 20, 21	H-12, H-21, H-22, H-32
18	175.8 C			
19	21.9 CH_3_	1.43 s	C: 1, 5, 9, 10	H-1, H-2, H-8, H-30
20	82.9 C			
21	26.6 CH_3_	1.40 s	C: 17, 20, 22	H-12, H-17, H-22
22	38.3 CH_2_	1.81 m		H-12, H-17, H-21
		1.66 m		
23	22.1 CH_2_	1.81 m		
		1.53 m		
24	37.8 CH_2_	1.99 m	C: 25, 26, 27	H-27
25	145.4 C			
26	110.3 CH_2_	4.78 brs	C: 24, 25, 27	H-27
27	22.2 CH_3_	1.70 s	C: 24, 25, 26	
30	16.5 CH_3_	1.11 s	C: 3, 4, 5, 31	H-2, H-6, H-19
31	27.9 CH_3_	1.31 s	C: 3, 4, 5, 30	H-3, H-5, H-6, H-1 Xyl1
32	20.5 CH_3_	0.92 s	C: 8, 13, 14, 15	H-7, H-12, H-15, H-17

**Table 4 marinedrugs-17-00358-t004:** ^13^C and ^1^H NMR chemical shifts and HMBC and ROESY correlations of carbohydrate moiety of psolusoside E (**2**). ^a^ Recorded at 176.04 MHz in C_5_D_5_N/D_2_O (4/1). ^b^ Bold is interglycosidic positions. ^c^ Italic is sulphate position. ^d^ Recorded at 700.13 MHz in C_5_D_5_N. Multiplicity by 1D TOCSY.

Atom	δ_C_ mult. ^a, b c^	δ_H_ mult. ^d^ (*J* in Hz)	HMBC	ROESY
Xyl1 (1→C-3)				
1	105.4 CH	4.77 d (7.4)	C-3	H-3; H-3, 5 Xyl1
2	**83.1** CH	4.00 t (8.8)	C: 1, 3 Xyl1; C: 1 Qui2	H-1 Qui2
3	77.6 CH	4.18 t (8.8)	C: 4 Xyl1	H-1 Xyl1
4	70.8 CH	4.12 m		
5	66.5 CH_2_	4.26 dd (5.4; 11.5)		
		3.61 t (10.9)		H-1, 3 Xyl1
Qui2 (1→2Xyl1)				
1	104.6 CH	5.12 d (7.7)	C: 2 Xyl1	H-2 Xyl1; H-3, 5 Qui2
2	75.7 CH	3.96 t (8.9)	C: 1, 3 Qui2	H-4 Qui2
3	75.1 CH	4.08 t (8.9)	C: 2, 4 Qui2	H-1, 5 Qui2
4	**87.7** CH	3.51 t (8.9)	C: 5 Qui2, 1 Glc3	H-1 Glc3; H-2 Qui2
5	71.3 CH	3.71 dd (5.9; 8.9)		H-1, 3 Qui2
6	17.8 CH_3_	1.67 d (5.9)	C: 4, 5 Qui2	H-4, 5 Qui2
Glc3 (1→4Qui2)				
1	104.9 CH	4.81 d (7.3)	C: 4 Qui2	H-4 Qui2; H-3,5 Glc3
2	73.2 CH	3.99 t (8.1)	C: 3 Glc3	H-4 Glc3
3	**87.3** CH	4.16 t (8.8)	C: 2, 4 Glc3; 1 MeGlc4	H-1 MeGlc4; H-1 Glc3
4	69.9 CH	3.88 t (8.8)	C: 3, 5, 6 Glc3	H-6 Glc3
5	75.0 CH	4.22 t (8.8)		H-1 Glc3
6	*67.6* CH_2_	5.16 brd (10.3)		
		4.77 t (8.8)		H-4 Glc3
MeGlc4(1→3Glc3)				
1	105.4 CH	5.22 d (7.3)	C: 3 Glc3	H-3 Glc3; H-3, 5 MeGlc4
2	74.8 CH	3.94 t (8.8)	C: 1, 3 MeGlc4	
3	87.8 CH	3.68 t (8.8)	C: 2, 4 MeGlc4, OMe	H-1, 5 MeGlc4; OMe
4	70.4 CH	4.11 t (8.8)	C: 3, 5, 6 MeGlc4	
5	78.2 CH	3.91 m	C: 3 MeGlc4	H-1, 3 MeGlc4
6	61.9 CH_2_	4.43 dd (2.6; 11.7)		
		4.24 dd (5.1; 11.7)		
OMe	60.5 CH_3_	3.85 s	C: 3 MeGlc4	

**Table 5 marinedrugs-17-00358-t005:** ^13^C and ^1^H NMR chemical shifts and HMBC and ROESY correlations of carbohydrate moiety of psolusoside F (**3**). ^a^ Recorded at 176.04 MHz in C_5_D_5_N/D_2_O (4/1). ^b^ Bold is interglycosidic positions. ^c^ Italic is sulphate position. ^d^ Recorded at 700.13 MHz in C_5_D_5_N. Multiplicity by 1D TOCSY.

Atom	δ_C_ mult. ^a,b,c^	δ_H_ mult. ^d^ (*J* in Hz)	HMBC	ROESY
Xyl1 (1→C-3)				
1	105.0 CH	4.71 d (7.3)	C: 3	H-3; H-3, 5 Xyl1
2	**83.1** CH	3.95 t (8.6)	C: 1, 3 Xyl1; C: 1 Qui2	H-1 Qui2
3	77.2 CH	4.15 t (8.6)	C: 2, 4 Xyl1	H-1, 5 Xyl1
4	70.2 CH	4.10 m		
5	66.0 CH_2_	4.25 dd (4.9; 11.6)	C: 3 Xyl1	
		3.63 t (11.0)		H-1 Xyl1
Qui2 (1→2Xyl1)				
1	104.8 CH	5.02 d (7.3)	C: 2 Xyl1	H-2 Xyl1; H-3, 5 Qui2
2	75.7 CH	3.94 t (8.7)	C: 1, 3 Qui2	
3	75.3 CH	3.99 t (8.7)		H-1 Qui2
4	**86.4** CH	3.58 t (8.7)	C: 3, 5 Qui2, 1 Glc3	H-1 Glc3
5	71.5 CH	3.67 dd (5.8; 10.2)		H-1 Qui2
6	17.9 CH_3_	1.65 d (5.8)		H-4, 5 Qui2
Glc3 (1→4Qui2)				
1	104.0 CH	4.88 d (8.2)	C: 4 Qui2	H-4 Qui2; H-3, 5 Glc3
2	73.5 CH	3.92 t (8.2)	C: 1, 3 Glc3	
3	**87.6** CH	4.13 t (8.8)	C: 4 Glc3; 1 MeGlc4	H-1 MeGlc4; H-1 Glc3
4	69.4 CH	3.84 t (8.8)	C: 3, 5 Glc3	
5	77.1 CH	3.93 m		
6	61.7 CH_2_	4.36 dd (2.9; 12.3)		
		4.03 dd (7.0; 12.3)		
MeGlc4 (1→3Glc3)				
1	104.8 CH	5.12 d (7.8)	C: 3 Glc3	H-3 Glc3; H-3, 5 MeGlc4
2	74.3 CH	3.79 t (9.4)	C: 1 MeGlc4	H-4 MeGlc4
3	86.4 CH	3.65 t (9.4)	C: 2, 4 MeGlc4, OMe	H-1 MeGlc4, OMe
4	69.9 CH	3.98 t (9.4)	C: 3, 5, 6 MeGlc4	H-6 MeGlc4
5	75.6 CH	4.04 dd (7.8; 10.9)		H-1 MeGlc4
6	*67.0* CH_2_	4.98 d (9.4)		
		4.74 dd (5.5; 10.9)		
OMe	60.5 CH_3_	3.76 s	C: 3 MeGlc4	

**Table 6 marinedrugs-17-00358-t006:** ^13^C and ^1^H NMR chemical shifts and HMBC and ROESY correlations of carbohydrate moiety of psolusoside G (**4**). ^a^ Recorded at 176.04 MHz in C_5_D_5_N/D_2_O (4/1). ^b^ Bold is interglycosidic positions. ^c^ Italic is sulphate position. ^d^ Recorded at 700.13 MHz in C_5_D_5_N. Multiplicity by 1D TOCSY.

Atom	δ_C_ mult. ^a,b,c^	δ_H_ mult. ^d^ (*J* in Hz)	HMBC	ROESY
Xyl1 (1→C-3)				
1	105.0 CH	4.72 d (7.5)	C: 3	H-3; H-3, 5 Xyl1
2	**81.8** CH	4.06 t (7.5)	C: 1, 3 Xyl1; C: 1 Glc2	H-1 Glc2
3	77.1 CH	4.16 t (8.8)	C: 2, 4 Xyl1	H-1, 5 Xyl1
4	70.1 CH	4.09 m		
5	66.0 CH_2_	4.23 dd (4.6; 10.9)		
		3.62 t (11.3)		H-1, 3 Xyl1
Glc2 (1→2Xyl1)				
1	104.2 CH	5.16 d (7.2)	C: 2 Xyl1	H-2 Xyl1; H-3, 5 Glc2
2	75.2 CH	3.93 t (9.5)	C: 1, 3 Glc2	
3	75.2 CH	4.02 t (9.5)	C: 2, 4 Glc2	H-1, 5 Glc2
4	**81.8** CH	3.95 t (9.5)	C: 3 Glc2, 1 Glc3	H-1 Glc3; H-6 Glc2
5	76.0 CH	3.72 m		H-1 Glc2
6	61.2 CH_2_	4.29 m		
Glc3 (1→4Glc2)				
1	103.8 CH	4.86 d (8.4)	C: 4 Glc2	H-4 Glc2; H-3 Glc3
2	73.3 CH	3.80 t (8.4)	C: 1 Glc3	
3	**86.5** CH	4.05 t (9.5)	C: 2, 4 Glc3; 1 MeGlc4	H-1 MeGlc4; H-1 Glc3
4	69.1 CH	3.76 t (9.5)	C: 5, 6 Glc3	
5	74.8 CH	4.04 m		
6	*67.4* CH_2_	5.95 dd (1.6; 10.3)		
		4.57 dd (5.4; 10.3)		
MeGlc4 (1→3Glc3)				
1	104.6 CH	5.07 d (7.6)	C: 3 Glc3	H-3 Glc3; H-3, 5 MeGlc4
2	74.2 CH	3.76 t (8.7)	C: 1 MeGlc4	
3	86.4 CH	3.61 t (8.2)	C: 2, 4 MeGlc4, OMe	H-1, 5 MeGlc4
4	69.8 CH	3.96 t (8.2)		
5	75.4 CH	3.99 m		H-1 MeGlc4
6	*67.1* CH_2_	4.93 d (10.8)		
		4.71 dd (5.4; 10.8)		
OMe	60.5 CH_3_	3.76 s	C: 3 MeGlc4	

**Table 7 marinedrugs-17-00358-t007:** ^13^C and ^1^H NMR chemical shifts and HMBC and ROESY correlations of carbohydrate moieties of psolusosides H (**5**) and H_1_ (**6**). ^a^ Recorded at 176.04 MHz in C_5_D_5_N. ^b^ Bold is interglycosidic positions. ^c^ Italic is sulphate position. ^d^ Recorded at 700.13 MHz in C_5_D_5_N. Multiplicity by 1D TOCSY.

Atom	δ_C_ mult. ^a,b,c^	δ_H_ mult. ^d^ (*J* in Hz)	HMBC	ROESY
Xyl1 (1→C-3)				
1	104.9 CH	4.73 d (7.5)	C-3	H-3; H-3, 5 Xyl1
2	**82.0** CH	4.07 t (8.0)	C: 1 Glc2; 1, 3 Xyl1	H-1 Glc2
3	77.2 CH	4.18 t (8.9)	C: 2, 4 Xyl1	H-1, 5 Xyl1
4	70.1 CH	4.11 m		
5	66.0 CH_2_	4.25 dd (5.6; 11.3)	C: 3, 4 Xyl1	
		3.63 dd (2.0; 11.2)		H-1, 3 Xyl1
Glc2 (1→2Xyl1)				
1	104.4 CH	5.19 d (6.8)	C: 2 Xyl1	H-2 Xyl1; H-3, 5 Glc2
2	75.2 CH	3.95 t (8.2)	C: 1 Glc2	
3	75.5 CH	4.05 t (8.7)	C: 2, 4 Glc2	H-1, 5 Glc2
4	**82.2** CH	3.97 t (8.7)	C: 1 Glc3; 3 Glc2	H-1 Glc3; H-6 Glc2
5	75.9 CH	3.77 m		H-1, 3 Glc2
6	61.6 CH_2_	4.33 m		H-4 Glc2
Glc3 (1→4Glc2)				
1	104.5 CH	4.88 d (7.9)	C: 4 Glc2	H-4 Glc2; H-3,5 Glc3
2	74.2 CH	3.83 t (9.0)	C: 1, 3 Glc3	H-4 Glc3
3	76.9 CH	4.10 t (9.0)	C: 2, 4 Glc3	H-1 Glc3
4	70.6 CH	3.97 t (9.0)	C: 5, 6 Glc3	H-2, 6 Glc3
5	75.7 CH	4.06 m	C: 4 Glc3	H-1 Glc3
6	*67.2* CH_2_	5.07 dd (2.8; 11.3)		
		4.73 dd (6.8; 11.3)	C: 5 Glc3	

**Table 8 marinedrugs-17-00358-t008:** ^13^C and ^1^H NMR chemical shifts and HMBC and ROESY correlations of aglycone moiety of psolusoside H (**5**). ^a^ Recorded at 176.04 MHz in C_5_D_5_N. ^b^ Recorded at 700.13 MHz in C_5_D_5_N/D_2_O (4/1).

Position	δ_C_ mult. ^a^	δ_H_ mult. ^b^ (*J* in Hz)	HMBC	ROESY
1	35.4 CH_2_	1.35 m		H-3, H-5, H-11, H-19
2	26.7 CH_2_	2.06 m		
		1.89 m		
3	89.2 CH	3.24 dd (3.8; 11.8)	C: 30, 1 Xyl1	H-1, H-5, H-31, H-1 Xyl1
4	39.2 C			
5	48.1 CH	0.92 dd (4.3; 11.6)		H-1, H-3, H-31
6	23.1 CH_2_	1.91 m		H-19, H-31
7	121.7 CH	5.63 m		H-15, H-32
8	143.9 C			
9	46.9 CH	3.54 brd (15.2)		H-19
10	35.7 C			
11	22.2 CH_2_	1.80 m		H-1
		1.53 m		H-32
12	29.4 CH_2_	2.19 brdd (5.8; 8.8)	C: 13, 18	H-17, H-21, H-32
13	56.6 C			
14	45.6 C			
15	51.8 CH_2_	2.65 d (15.9)	C: 13, 16, 32	H-7, H-32
		2.32 d (16.1)	C: 14, 16, 32	H-7
16	213.8 C			
17	63.3 CH	2.87 s	C: 12, 13, 16, 18, 20, 21	H-12, H-21, H-22, H-32
18	179.0 C			
19	23.8 CH_3_	1.10 s	C: 1, 9, 10	H-1, H-2, H-6, H-9
20	83.6 C			
21	26.0 CH_3_	1.45 s	C: 17, 20, 22	H-12, H-17, H-22
22	38.1 CH_2_	1.71 m		H-17, H-21
		1.56 m		
23	22.0 CH_2_	1.71 m		
		1.43 m		
24	37.7 CH_2_	1.90 m	C: 25, 26	H-26
25	145.4 C			
26	110.3 CH_2_	4.70 brs	C: 24, 27	H-27
		4.69 brs	C: 24, 27	H-27
27	22.0 CH_3_	1.63 s	C: 24, 25, 26	
30	17.1 CH_3_	1.07 s	C: 3, 4, 5, 31	H-2, H-6, H-6 Glc2
31	28.5 CH_3_	1.20 s	C: 3, 4, 5, 30	H-3, H-5, H-6, H-1 Xyl1
32	31.7 CH_3_	1.16 s	C: 8, 13, 14, 15	H-7, H-11, H-12, H-15, H-17

**Table 9 marinedrugs-17-00358-t009:** ^13^C and ^1^H NMR chemical shifts and HMBC and ROESY correlations of aglycone moiety of psolusoside H_1_ (**6**). ^a^ Recorded at 176.04 MHz in C_5_D_5_N. ^b^ Recorded at 700.13 MHz in C_5_D_5_N/D_2_O (4/1).

Position	δ_C_ mult. ^a^	δ_H_ mult. ^b^ (*J* in Hz)	HMBC	ROESY
1	36.0 CH_2_	1.34 m		H-3, H-11
2	26.8 CH_2_	2.03 m		
		1.88 m		H-30
3	89.5 CH	3.24 dd (4.0; 11.7)	C: 30, 31, 1 Xyl1	H-1, H-5, H-31, H-1 Xyl1
4	39.3 C			
5	48.0 CH	0.93 dd (4.4; 10.8)	C: 4, 10, 19, 30, 31	H-3, H-31
6	23.1 CH_2_	1.91 m		H-31
7	119.8 CH	5.62 m		H-15, H-32
8	146.5 C			
9	47.2 CH	3.37 brd (14.3)		H-19
10	35.4 C			
11	22.7 CH_2_	1.70 m		H-1
		1.49 m		
12	30.3 CH_2_	2.00 m	C: 13, 18	H-32
13	58.6 C			
14	51.6 C			
15	34.1 CH_2_	1.76 m	C: 13	H-7
		1.50 m		H-32
16	24.5 CH_2_	1.97 m		H-32
		1.84 m		
17	52.9 CH	2.29 dd (4.0; 10.5)	C: 13, 18	H-12, H-32
18	181.0 C			
19	23.8 CH_3_	1.08 s	C: 1, 5, 9, 10	H-2, H-9
20	84.7 C			
21	22.7 CH_3_	1.28 s	C: 17, 20, 22	H-23
22	40.7 CH_2_	1.71 m		
		1.55 m		H-24
23	22.0 CH_2_	1.47 m	C: 22, 24	H-21
24	37.7 CH_2_	1.99 m	C: 22, 23, 25, 26	H-22
		1.88 m		
25	145.5 C			
26	110.6 CH_2_	4.78 brs	C: 24, 27	H-27
		4.74 brs	C: 24, 27	H-24, H-27
27	22.1 CH_3_	1.66 s	C: 24, 25, 26	
30	17.2 CH_3_	1.06 s	C: 3, 4, 5, 31	H-2
31	28.6 CH_3_	1.19 s	C: 3, 4, 5, 30	H-3, H-5, H-6, H-1 Xyl1
32	30.7 CH_3_	1.11 s	C: 8, 13, 14, 15	H-7, H-11, H-12, H-15, H-17

**Table 10 marinedrugs-17-00358-t010:** ^13^C and ^1^H NMR chemical shifts and HMBC and ROESY correlations of carbohydrate moiety of psolusoside I (**7**). ^a^ Recorded at 176.04 MHz in C_5_D_5_N/D_2_O (4/1). ^b^ Bold is interglycosidic positions. ^c^ Italic is sulphate position. ^d^ Recorded at 700.13 MHz in C_5_D_5_N. Multiplicity by 1D TOCSY.

Atom	δ_C_ mult. ^a,b,c^	δ_H_ mult. ^d^ (*J* in Hz)	HMBC	ROESY
Xyl1 (1→C-3)				
1	104.4 CH	4.63 d (7.3)	C: 3	H-3; H-3, 5 Xyl1
2	**83.2** CH	3.75 t (7.3)	C: 3 Xyl1; C: 1 Xyl2	H-1 Xyl2; H-4 Xyl1
3	75.2 CH	3.99 t (6.7)	C: 2, 4 Xyl1	H-1, 5 Xyl1
4	**80.2** CH	3.99 t (6.7)	C: 3 Xyl1	
5	63.5 CH_2_	4.35 dd (4.9; 11.0)	C: 1, 3, 4 Xyl1	
		3.60 t (9.2)		H-1, 3 Xyl1
Xyl2 (1→2Xyl1)				
1	105.6 CH	4.79 d (7.6)	C: 2 Xyl1	H-2 Xyl1; H-3, 5 Xyl2
2	75.0 CH	3.91 t (9.3)	C: 1, 3 Xyl2	
3	75.0 CH	4.06 t (9.3)	C: 2, 4 Xyl2	H-1, 5 Xyl2
4	**79.9** CH	3.88 m	C: 1 Glc3	H-1 Glc3
5	64.5 CH_2_	4.24 dd (5.1; 11.0)	C: 1, 3, 4 Xyl2	
		3.45 t (9.3)		H-1, 3 Xyl2
Glc3 (1→4Xyl2)				
1	103.9 CH	4.71 d (8.0)	C: 4 Xyl2	H-4 Xyl2; H-3, 5 Glc3
2	73.8 CH	3.75 t (8.0)	C: 1, 3 Glc3	H-4 Glc3
3	76.8 CH	4.08 t (9.3)	C: 2, 4 Glc3	H-1 Glc3
4	70.6 CH	3.84 t (9.3)	C: 3, 5, 6 Glc3	
5	75.1 CH	4.03 t (8.7)		
6	*67.6* CH_2_	4.97 d (10.7)		
		4.63 dd (6.7; 10.7)	C: 5 Glc3	
Glc4 (1→4Xyl1)				
1	103.9 CH	4.82 d (8.1)	C: 4 Xyl1	H-4 Xyl1; H-3, 5 Glc4
2	73.9 CH	3.82 t (8.1)	C: 1, 3 Glc4	
3	76.8 CH	4.10 t (9.0)	C: 2, 4 Glc4	H-1 Glc4
4	71.0 CH	3.86 t (9.0)	C: 3, 5, 6 Glc4	
5	75.1 CH	4.11 m		H-1 Glc4
6	*67.9* CH_2_	5.02 d (9.0)		
		4.57 dd (7.8; 11.2)	C: 5 Glc4	

**Table 11 marinedrugs-17-00358-t011:** ^13^C and ^1^H NMR chemical shifts and HMBC and ROESY correlations of aglycone moiety of psolusoside I (**7**). ^a^ Recorded at 176.04 MHz in C_5_D_5_N/D_2_O (4/1). ^b^ Recorded at 700.13 MHz in C_5_D_5_N/D_2_O (4/1).

Position	δ_C_ mult. ^a^	δ_H_ mult. ^b^ (*J* in Hz)	HMBC	ROESY
1	35.8 CH_2_	1.32 m		H-3, H-11, H-31
2	26.8 CH_2_	1.98 m		
		1.82 m		H-19, H-30
3	88.9 CH	3.19 dd (3.9; 11.6)	C: 4, 30, 31	H-1, H-5, H-31, H-1 Xyl1
4	39.2 C			
5	47.9 CH	0.90 dd (4.5; 11.4)	C: 4, 10, 19, 30, 31	H-3, H-31
6	23.1 CH_2_	1.94 m		H-19, H-30
7	120.2 CH	5.60 m		H-15, H-32
8	145.6 C			
9	47.0 CH	3.32 brd (13.8)		H-19
10	35.4 C			
11	22.4 CH_2_	1.71 m		H-1
		1.46 m		H-32
12	31.2 CH_2_	2.10 m	C: 13, 14, 18	H-12, H-17, H-32
13	59.3 C			
14	47.3 C			
15	43.5 CH_2_	2.55 dd (7.3; 12.0)	C: 13, 14, 17, 32	H-7, H-32
		1.62 dd (9.0; 12.0)	C: 14, 16, 32	
16	75.2 CH	5.82 brq (8.6)	OAc	H-32
17	54.5 CH	2.67 d (9.1)	C: 12, 13, 18, 21	H-12, H-21, H-32
18	180.2 C			
19	23.8 CH_3_	1.11 s	C: 1, 5, 9, 10	H-1, H-2, H-6, H-9
20	85.5 C			
21	28.0 CH_3_	1.52 s	C: 17, 20, 22	H-12, H-17, H-22
22	38.1 CH_2_	1.82 dt (3.6; 12.7)	C: 20, 23	H-21
		1.25 dt (4.5; 12.7)	C: 20, 23	
23	22.9 CH_2_	1.47 m	C: 22, 24	
		1.36 m		
24	38.1 CH_2_	1.92 m	C: 22, 23, 25, 26	H-26
		1.82 m		
25	145.4 C			
26	110.8 CH_2_	4.73 brs	C: 24, 27	H-27
		4.72 brs		
27	22.0 CH_3_	1.65 s	C: 24, 25, 26	
30	16.7 CH_3_	0.94 s	C: 3, 4, 5, 31	H-2, H-6
31	28.2 CH_3_	1.12 s	C: 3, 4, 5, 30	H-3, H-5, H-6, H-1 Xyl1
32	32.1 CH_3_	1.15 s	C: 8, 13, 14, 15	H-7, H-11, H-12, H-15, H-17
OAc	170.7 C			
	21.2 CH_3_	2.02 s	OAc	

**Table 12 marinedrugs-17-00358-t012:** Hemolytic activities of glycosides **2**–**7** and psolusoside A against mouse erythrocytes, cytotoxic activity against the ascite form of mouse Ehrlich carcinoma cells, mouse neuroblasoma Neuro 2A cells, and normal epithelial JB-6 cells.

Glycoside	Cytotoxicity EC_50_, µМ			
	Erhythrocytes	Ehrlich carcinoma	Neuro-2A	JB-6
Psolusoside E (**2**)	0.23	55.75	3.96	-
Psolusoside F (**3**)	2.8	96.2	10.8	23.8
Psolusoside G (**4**)	4.2	98.3	7.3	>100.0
Psolusoside H (**5**)	2.5	>100.0	47.5	43.2
Psolusoside H_1_ (**6**)	2.7	92.5	10.7	32.3
Psolusoside I (**7**)	18.3	87.4	26.8	37.2
Psolusoside A	1.4	30.9	2.9	7.5

**Table 13 marinedrugs-17-00358-t013:** Cytotoxic activities of psolusosides A, B (**1**), E (**2**), F (**3**), H (**5**), H_1_ (**6**), and I (**7**) against the HT-29 cells and ability for inhibition of their colony.

Glycoside	MTS Assay	Soft Agar Assay
	IC_50_	ICCF_50_, μM
psolusoside B (**1**)	>20 μM	>10
psolusoside E (**2**)	>20 μM	0.1±0.03
psolusoside F (**3**)	>20 μM	0.5±0.03
psolusoside H (**5**)	>20 μM	>10
psolusoside H_1_ (**6**)	>20 μM	9±0.3
psolusoside I (**7**)	>20 μM	10±0.4
psolusoside A	>20 μM	>10

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
