# Peer review of "Structures and Bioactivities of Six New Triterpene Glycosides, Psolusosides E, F, G, H, H1, and I and the Corrected Structure of Psolusoside B from the Sea Cucumber Psolus fabricii"

_marinedrugs, 2019, doi:10.3390/md17060358_

Reviewer 1 Report

Marinedrugs-524666

Structures and bioactivities of five new triterpene glycosides, psolusosides E, F, G, H, H1, I and the corrected structure of psolusoside B from the sea cucumber Psolus fabricii

This manuscript describes isolation and structure elucidation of six novel saponines (2-6), structure revision of one known saponine (1), and biological activities of these molecules together with one known analgue.

Structure determination part was precisely and exhaustively described based on the NMR spectral data and comparison to the literature data. In the previous report, molecular formula of compound 1 was determined by HR-FABMS analysis resulted in miss-assignment of the de-sulfated fragment as a molecular-ion. But, in this study, authors used ESI-MS system which is milder ionization method enable to obtain correct molecular formula resulted in revision of the proposed structure of 1. Judging from the database search, all proposed compounds are novel, especially aglycon part of 1 is rare.

In the biological activity part, authors assessed variety of activities including hemolysis, cytotoxicity against several cell lines, colony formation activity, and synergistic effect upon radioactive irradiation. Obtained triterpene glycoside showed potent hemolytic activities and some structure-activity relationship information was also acquired.

Considering the above results, quality and quantity of this manuscript is more than standard of the natural products isolation and structure elucidation studies. To increase the value of this manuscript, and also convenience for the readers, authors should provide graphical summary of speculated biosynthetic scheme, and also graphical review of the structure-activity relationship data.

Authors also should check the title of the manuscript, the number of new compounds may be six not five.

Author Response

1)      We have corrected the title where the number of new natural products was increased by six.

2)      We have provided supplementary materials with all necessary NMR information.

3)      We have inserted two schemes of biosynthesis.

4)      The construction of graphical scheme of structure / activity relations (SAR) seems to be not necessary in this article because it may be a subject of independent article where we intend to use molecular modelling approach for explanation of the SAR.

Reviewer 2 Report

The manuscript is reported six new holostane type triterpenoid glucosides, and structural revision of psolusoside B with study of biological activities. Detailed of NMR spectral data were analyzed well together with ESI-MS/MS analysis. Also in many 1H NMR data, coupling constant values have small discrepancy, some typical one is mentioned separately, but there are many points to be improved. Please add all NMR spectra into Supplemental material, then it will be clear to check them. Other minor comments are listed below. 

L2 Five => Six 

L74 Numbering  30-> 29, 31-> 28, 32 -> 30  

I know that such a numbering is used in many literature, but why is it different from ordinary lanostane numbering system?

For example. Tetrahedron 2004, 60, 2987-2992.

If those methyl group is oxidized, its naming will be confused i.e. Where is 30-hydroxy?

L85, L455 [M2Na + Na]2+  =>  [M2Na + 2Na]2+

L146 Table 2 Coupling constant values are not matched each other in many cases.

For example, Xyl H-2 is (t, 8.0) would be (dd 8.8, 7.3).

In many tables, many coupling constant values are not matched.

L175 Table 4 

Xyl1 Atom 1 66.5 CH2 => 66.5 CH2

L271 Geminal coupling constant should be matched. Average value of 16.0Hz is acceptable. 

L414 Please add information of ESI-MS apparatus in negative mode.

L618 147-185 => 147

K620 317, 1-35 => 317

Author Response

1)      We have corrected the title of the manuscript and increased the number of new compounds by six.

2)      Concerning the numbering in the lanostane cyclic system. We have used the current system because it is traditional for the articles in this field. It seems to be necessary for comparability of spectral data. This numbering system is very old and based on and erroneous structure of lanostane derivatives but it became a traditional system. Hence there is no necessary to change something in order to avoid the confusion of the readers and ourselves. In any cases we provide such numbering in each our article and there is no a possibility for any confusion.

3)      We have fixed the formulae of two-charged cations on M2Na + 2Na at the line 85 and 455

4)      The referee notes:

Coupling constant values are not matched each other in many cases. For example, Xyl H-2 is (t, 8.0) would be (dd 8.8, 7.3). In many tables, many coupling constant values are not matched.

The reply: the corresponding signal is strong overlapped doublet of doublets and really looks as asymmetric triplet. In such and similar situations we just describe the visible signal multiplicity and one of the corresponding constants and not a more. We also presented an original spectral information in Supplementary Materials.

             5)      CH2 in the Table 4 is fixed.

6)      Referee Note: L271 Geminal coupling constant should be matched. Average value of 16.0Hz is accetable. Reply: we agree and the corresponding values are fixed.

7)      ESI-MS apparatus in negative mode is the same as in positive mode. We have corrected the text.

8)      The references [2] and [3] are fixed along with the referee recommendations.

9)      The supplementary materials with all the NMR information are added.

Reviewer 3 Report

The present manuscript reports the isolation, structural determination and biological activities of five new triterpene glycosides, namely psolusosides E, F, G, H and I and the structural revision of a known related glycoside, psolusoside B. In my opinion, this work contains a thorough study of structural determination, which has been correctly rationalized and justified by techniques of NMR spectroscopy and mass spectrometry. However, given the nature of the study, the authors must provide all NMR spectra corresponding to all the experiments (mono- and bidimentional experiments) carried out to support the structural elucidation described in the manuscript. These NMR experiments can be provided as Supplementary Information, and, for sure, will be of great interest and  usefulness for researchers involved in the chemistry and biology of triterpene glycosides.

Author Response

The NMR spectral information is provided as supplementary materials.

Round  2

Reviewer 3 Report

The supplementary information provided by the authors is correct and represents a valuable information for the readership involved in this field. However, the authors must complete the information, indicating deuterated solvent and frequency in which the NMR spectra were recorded. This information can be annotated in the legends of the figures.

Author Response

We have indicated the deuterated solvents and frequencies in which the NMR spectra were recorded in the legends of the figures in Supplementary data. We have also added the 13C NMR spectra of compounds 1 - 7 and content page to Supplementary data.
